# DATeS: A Highly-Extensible Data Assimilation Testing Suite v1.0

Ahmed Attia[1] and Adrian Sandu[2]

[1]Mathematics and Computer Science Division , Argonne National Laboratory , 9700 S. Cass Ave. Bldg. 240, Lemont, IL 60439, USA. , E-mail: attia@mcs.anl.gov

[2]Computational Science Laboratory , Department of Computer Science , Virginia Polytechnic Institute and State University , 2201 Knowledgeworks II, 2202 Kraft Drive, Blacksburg, VA 24060, USA , E-mail: sandu@cs.vt.edu

*Correspondence to:* Ahmed Attia (attia@mcs.anl.gov)

**Abstract.** A flexible and highly-extensible data assimilation testing suite, named DATeS, is described in this paper. DATeS aims to offer a unified testing environment that allows researchers to compare different data assimilation methodologies and understand their performance in various settings. The core of DATeS is implemented in Python and takes advantage of its object-oriented capabilities. The main components of the package (the numerical models, the data assimilation algorithms, the linear algebra solvers, and the time discretization routines) are independent of each other, which offers great flexibility to configure data assimilation applications. DATeS can interface easily with large third-party numerical models written in Fortran or in C, and with a plethora of external solvers.

## 1 Introduction

Data Assimilation (DA) refers to the fusion of information from different sources, including priors, predictions of a numerical model, and snapshots of reality, in order to produce accurate description of the state of a physical system of interest (Daley, 1991; Kalnay, 2002). DA research is of increasing interest for a wide range of fields including geoscience, numerical weather forecasts, atmospheric composition predictions, oil reservoir simulations, and hydrology. Two approaches have gained wide popularity for solving the DA problems, namely ensemble and variational approaches. The ensemble approach is rooted in statistical estimation theory and uses an ensemble of states to represent the underlying probability distributions. The variational approach, rooted in control theory, involves solving an optimization problem to obtain a single "*analysis*" as an estimate of the true state of the system of concern. The variational approach does not provide an inherent description of the uncertainty associated with the obtained analysis, however it is less sensitive to physical imbalances prevalent in the ensemble approach. Hybrid methodologies designed to harnesses the best of the two worlds are an on-going research topic.

Numerical experiments are an essential ingredient in the development of new DA algorithms. Implementation of numerical experiments for DA involves linear algebra routines, a numerical model along with time integration routines, and an assimilation algorithm. Currently available testing environments for DA applications are either very simplistic or very general, many are tied to specific models, and are usually completely written in a specific language. A researcher who wants to test a new algorithm with different numerical models written in different languages might have to re-implement his/her algorithm using

the specific settings of each model. A unified testing environment for DA is important to enable researchers to explore different aspects of various filtering and smoothing algorithms with minimal coding effort.

The DA Research Section (DAReS) at the National Center for Atmospheric Research (NCAR) provides DART (Anderson et al., 2009) as a community facility for ensemble filtering. The DART platform is currently the gold standard for ensemble-based Kalman filtering algorithm implementations. It is widely used in both research and operational settings, and interfaces to most important geophysical numerical models are available. DART employs a modular programming approach and adheres strictly to solid software engineering principles. DART has a long history, and is continuously well maintained; new ensemble-based Kalman filtering algorithms that appear in the literature are routinely added to its library. Moreover it gives access to practical, and well-established parallel algorithms. DART is, by design, very general in order to support operational settings with many types of geophysical models. Using DART requires a non-trivial learning overhead. The fact that DART is mainly written in Fortran makes it a very efficient testing platform, however this limits to some extent the ability to easily employ third party implementations of various components.

Matlab programs are often used to test new algorithmic ideas due to its ease of implementation. A popular set of Matlab tools for ensemble-based DA algorithms is provided by the Nansen Environmental and Remote Sensing Center (NERSC), with the code available from (Evensen and Sakov, 2009). A Matlab toolbox for uncertainty quantification (UQ) is UQLab (Marelli and Sudret, 2014). Also, for the newcomers to the DA filed, a concise set of Matlab codes is provided through the pedagogical applied mathematics reference (Law et al., 2015). Matlab is generally a very useful environment for small-to-medium scale numerical experiments.

Python is a modern high-level programming language that gives the power of reusing existing pieces of code via inheritance, and thus its code is highly-extensible. Moreover, it is a powerful scripting tool for scientific applications that can be used to glue legacy codes. This can be achieved by writing wrappers that can act as interfaces. Building wrappers around existing C, and Fortran code is a common practice in scientific research. Several automatic wrapper generation tools, such as SWIG (Beazley et al., 1996) and F2PY (Peterson, 2009), are available to create proper interfaces between Python and lower level languages. While translating Matlab code to Python is a relatively easy task, one can call Matlab functions from Python using the Matlab Engine API. Moreover, Unlike Matlab, Python is freely available on virtually all Linux, MacOS, and Windows platforms, and therefore Python software is easily accessible and has excellent portability. When using Python, instead of Fortran or C, one generally trades some computational performance for programming productivity. The performance penalty in the scientific calculations is minimized by delegating computationally intensive tasks to compiled languages such as Fortran. This approach is followed by the scientific computing Python modules Numpy and Scipy, which enables writing computationally efficient scientific Python code. Moreover, Python is one of the easiest programming languages to learn, even without background knowledge about programming.

This paper presents a highly-extensible Python-based DA testing suite. The package is named DATeS, and is intended to be an *open-source, extendable* package positioned between the simple typical research-grade implementations and the professional implementation of DART, but with the capability to utilize large physical models. Researchers can use it as experimental testing pad where they can focus on coding only their new ideas without worrying much about the other pieces of the DA

process. Moreover, DATeS can be effectively used for educational purposes where students can use it as an interactive learning tool for DA applications. The code developed by a researcher in the DATeS framework should fit with all other pieces in the package with minimal-to-no effort, as long as the programmer follows the "*flexible*" rules of DATeS. As an initial illustration of its capabilities DATeS has been used to implement and carry out the numerical experiments in (Attia et al., 2018; Moosavi
et al., 2018; Attia and Constantinescu, 2018).

The paper is structured as follows. Section 2 reviews the DA problem and the most widely used approaches to solve it. Section 3 describes the architecture of the DATeS package. Section 4 takes a user-centric and example-based approach for explaining how to work with DATeS, and Section 5 demonstrates the main guidelines of contributing to DATeS. Conclusions and future development directions are discussed in Section 6.

## 2  Data Assimilation

This section gives a brief overview of the basic discrete-time formulations of both statistical and variational DA approaches. The formulation here is far from conclusive, and is intended only as a quick review. For detailed discussions on the various DA mathematical formulations and algorithms; see e.g., (Asch et al., 2016; Evensen, 2009; Law et al., 2015).

The main goal of a DA algorithm is to give an accurate representation of the "unknown" true state $\mathbf{x}^{\text{true}}(t_k)$ of a physical
system, at a specific time instant $t_k$. Assuming $\mathbf{x}_k \in \mathbb{R}^{\text{N}_{\text{state}}}$ is a discretized approximation of $\mathbf{x}^{\text{true}}(t_k)$, the time-evolution of the physical system over the time interval $[t_k, t_{k+1}]$ is approximated by the discretized forward model:

$$\mathbf{x}_{k+1} = \mathcal{M}_{k,\,k+1}(\mathbf{x}_k), \quad k = 0, 1, \ldots, N-1. \tag{1}$$

The model-based simulations, represented by the model states, are inaccurate and must be corrected given noisy measurements $\mathbf{Y}$ of the physical system. Since the model state and observations are both contaminated with errors, a probabilistic
formulation is generally followed. The prior distribution $\mathcal{P}^{\text{b}}(\mathbf{x}_k)$ encapsulates the knowledge about the model state at time instant $t_k$ before additional information is incorporated. The likelihood function $\mathcal{P}(\mathbf{Y}|\mathbf{x}_k)$ quantifies the deviation of the prediction of model observations from the collected measurements. The corrected knowledge about the system, is described by the posterior distribution formulated by applying Bayes' theorem:e

$$\mathcal{P}^{\text{a}}(\mathbf{x}_k|\mathbf{Y}) = \frac{\mathcal{P}^{\text{b}}(\mathbf{x}_k)\,\mathcal{P}(\mathbf{Y}|\mathbf{x}_k)}{\mathcal{P}(\mathbf{Y})} \propto \mathcal{P}^{\text{b}}(\mathbf{x}_k)\,\mathcal{P}(\mathbf{Y}|\mathbf{x}_k), \tag{2}$$

where $\mathbf{Y}$ refers to the data (observations) to be assimilated. In the sequential filtering context $\mathbf{Y}$ is a single observation, while in the smoothing context, it generally stands for several observations $\{\mathbf{y}_1, \ldots, \mathbf{y}_m\}$ to be assimilated simultaneously.

In the so-called "Gaussian framework", the prior is assumed to be Gaussian $\mathcal{N}(\mathbf{x}_k^{\text{b}}, \mathbf{B}_k)$ where $\mathbf{x}_k^{\text{b}}$ is a prior state, e.g. a model-based forecast, and $\mathbf{B}_k \in \mathbb{R}^{\text{N}_{\text{state}} \times \text{N}_{\text{state}}}$ is the prior covariance matrix. Moreover, the observation errors are assumed to be Gaussian $\mathcal{N}(0, \mathbf{R}_k)$, with $\mathbf{R}_k \in \mathbb{R}^{\text{N}_{\text{obs}} \times \text{N}_{\text{obs}}}$ being the observation error covariance matrix at time instant $t_k$, and observation
errors are assumed to be uncorrelated from background errors. In practical applications, the dimension of the observation space is much less than the state space dimension, that is $\text{N}_{\text{obs}} \ll \text{N}_{\text{state}}$.

Consider assimilating information available about the system state at time instance $t_k$, the posterior distribution follows from (2) as:

$$\mathcal{P}^{\mathrm{a}}(\mathbf{x}_k|\mathbf{y}_k) \propto \mathcal{P}^{\mathrm{b}}(\mathbf{x}_k)\mathcal{P}(\mathbf{y}_k|\mathbf{x}_k) \propto \exp\left(-\mathcal{J}(\mathbf{x}_k)\right),$$

$$\mathcal{J}(\mathbf{x}_k) = \frac{1}{2}\|\mathbf{x}_k - \mathbf{x}_k^{\mathrm{b}}\|^2_{\mathbf{B}_k^{-1}} + \frac{1}{2}\|\mathbf{y}_k - \mathcal{H}_k(\mathbf{x}_k)\|^2_{\mathbf{R}_k^{-1}}. \tag{3}$$

where the scaling factor $\mathcal{P}(\mathbf{y}_k)$ is dropped. Here, $\mathcal{H}_k$ is an observation operator that maps a model state $\mathbf{x}_k$ into the observation space.

Applying (2) or (3), in large-scale settings, even under the simplified Gaussian assumption, is not computationally feasible. In practice, a Monte-Carlo approach is usually followed. Specifically, ensemble-based sequential filtering methods such as ensemble Kalman filter (EnKF) (Tippett et al., 2003; Whitaker and Hamill, 2002; Burgers et al., 1998; Houtekamer and Mitchell, 1998; Zupanski et al., 2008; Sakov et al., 2012; Evensen, 2003; Hamill and Whitaker, 2001; Evensen, 1994; Houtekamer and Mitchell, 2001; Smith, 2007), and maximum likelihood ensemble filter (MLEF) (Zupanski, 2005) use ensembles of states to represent the prior, and the posterior distribution. A prior ensemble $\mathbf{X}_k = \{\mathbf{x}(e)\}_{e=1,2,\ldots,\mathrm{N_{ens}}}$, approximating the prior distributions, is obtained by propagating analysis states from a previous assimilation cycle at time $t_{k-1}$ by applying (1). Most of the ensemble based DA methodologies work by transforming the prior ensemble into an ensemble of states collected from the posterior distribution, namely the analysis ensemble. The transformation in the EnKF framework is applied following the update equations of the well-known Kalman filter (Kalman and Bucy, 1961; Kalman, 1960). An estimate of the true state of the system, i.e. the analysis, is obtained by averaging the analysis ensemble, while the posterior covariance is approximated by the covariance matrix of the analysis ensemble.

The maximum a posteriori (MAP) estimate of the true state is the state that maximizes the posterior probability density function (PDF). Alternatively the MAP estimate is the minimizer of the negative logarithm (negative-log) of the posterior PDF. The MAP estimate can be obtained by solving the following optimization problem:

$$\min_{\mathbf{x}_k} \mathcal{J}(\mathbf{x}_k) = \frac{1}{2}\left\|\mathbf{x}_k - \mathbf{x}_k^{\mathrm{b}}\right\|^2_{\mathbf{B}_k^{-1}} + \left\|\mathbf{y}_k - \mathcal{H}_k(\mathbf{x}_k)\right\|^2_{\mathbf{R}_k^{-1}}. \tag{4}$$

This formulates the three-dimensional variational (3D-Var) DA problem. Derivative-based optimization algorithms used to solve (4) require the derivative of the negative-log of the posterior PDF (4):

$$\nabla_{\mathbf{x}_k}\mathcal{J}(\mathbf{x}_k) = \mathbf{B}_k^{-1}\left(\mathbf{x}_k - \mathbf{x}_k^{\mathrm{b}}\right) + \mathbf{H}_k^T\mathbf{R}_k^{-1}\left(\mathbf{y}_k - \mathcal{H}_k(\mathbf{x}_k)\right), \tag{5}$$

where $\mathbf{H}_k = \partial\mathcal{H}_k/\partial\mathbf{x}_k$ is the sensitivity (e.g. the Jacobian) of the observation operator $\mathcal{H}_k$ evaluated at $\mathbf{x}_k$. Unlike ensemble filtering algorithms, the optimal solution of (4) provides a single estimate of the true state, and does not provide a direct estimate of associated uncertainty.

Assimilating several observations $\mathbf{Y} := \{\mathbf{y}_0, \mathbf{y}_1, \ldots, \mathbf{y}_m\}$ simultaneously requires adding time as a fourth dimension to the DA problem. Let $\mathcal{P}^{\mathrm{b}}(\mathbf{x}_0)$ be the prior distribution of the system state at the beginning of a time window $[t_0, t_F]$ over which the observations are distributed. Assuming the observations' errors are temporally uncorrelated, the posterior distribution of the

system state at the initial time of the assimilation window $t_0$ follows by applying Equation (2) as:

$$\mathcal{P}^{\mathrm{a}}(\mathbf{x}_0) \propto \mathcal{P}^{\mathrm{b}}(\mathbf{x}_0)\,\mathcal{P}(\mathbf{y}_0, \mathbf{y}_1, \ldots, \mathbf{y}_m | \mathbf{x}_0) \propto \exp\left(-\mathcal{J}(\mathbf{x}_0)\right),$$

$$\mathcal{J}(\mathbf{x}_0) = \frac{1}{2}\left\|\mathbf{x}_0 - \mathbf{x}_0^{\mathrm{b}}\right\|^2_{\mathbf{B}_0^{-1}} + \frac{1}{2}\sum_{k=0}^{m}\left\|\mathbf{y}_k - \mathcal{H}_k(\mathbf{x}_k)\right\|^2_{\mathbf{R}_k^{-1}}.$$

(6)

In the statistical approach, ensemble-based smoothers such as the ensemble Kalman smoother (EnKS) are used to approximate the posterior (6) based on an ensemble of states. Similar to the ensemble filters, the analysis ensemble generated by a smoothing
algorithm can be used to provide an estimate of the posterior first-order moment. It also can be used to provide a flow-dependent ensemble covariance matrix to approximate the posterior true second order moment.

The MAP estimate of the true state at the initial time of the assimilation window can be obtained by solving the following optimization problem:

$$\min_{\mathbf{x}_0} \mathcal{J}(\mathbf{x}_0) = \frac{1}{2}\left\|\mathbf{x}_0 - \mathbf{x}_0^{\mathrm{b}}\right\|^2_{\mathbf{B}_0^{-1}} + \frac{1}{2}\sum_{k=0}^{m}\left\|\mathbf{y}_k - \mathcal{H}_k(\mathbf{x}_k)\right\|^2_{\mathbf{R}_k^{-1}}.$$

(7)

This is the standard formulation of the four-dimensional variational (4D-Var) DA problem. The solution of the 4D-Var problem is equivalent to the MAP of the smoothing posterior in the Gaussian framework. The Jacobian of the (7) with respect to the model state at the initial time of the assimilation window reads

$$\nabla_{\mathbf{x}_0} \mathcal{J}(\mathbf{x}_0) = \mathbf{B}_0^{-1}(\mathbf{x}_0 - \mathbf{x}_0^{\mathrm{b}}) + \sum_{k=0}^{m} \mathbf{M}_{0,k}^{T}\mathbf{H}_k^{T}\mathbf{R}_k^{-1}\left(\mathcal{H}_k(\mathbf{x}_k) - \mathbf{y}_k\right),$$

(8)

where $\mathbf{M}_{0,k}^{T}$ is the adjoint of the tangent linear model operator, and $\mathbf{H}_k^{T}$ is the adjoint of the observation operator sensitiv-
ity. Similar to the 3D-Var case (4), the solution of Equation (7) provides a single best estimate (the analysis) of the system state without providing consistent description of the uncertainty associated with this estimate. The variational problem (7) is referred to as strong-constraint formulation, where a perfect-model approach is considered. In the presence of model errors, an additional term is added, resulting in a weak-constraint formulation. A general practice is to assume that the model errors follow a Gaussian distribution $\mathcal{N}(\mathbf{0}, \mathbf{Q}_k)$, with $\mathbf{Q}_k \in \mathbb{R}^{\mathrm{N_{state}} \times \mathrm{N_{state}}}$ being the model error covariance matrix at time instant $t_k$.
In non-perfect-model settings, an additional term characterizing state deviations is added to the variational objectives (4, 7). The model error term depends on the approach taken to solve the weak-constraint problem, and usually involves the model error probability distribution.

In idealized settings, where the model is linear, the observation operator is linear, and the underlying probability distributions are Gaussian, the posterior is also Gaussian, however this is rarely the case in real applications. In nonlinear or non-Gaussian
settings, the ultimate objective of a DA algorithm is to sample all probability modes of the posterior distribution, rather than just producing a single estimate of the true state. Algorithms capable of accommodating non-Gaussianity are too limited and have not been successfully tested in large-scale settings.

Particle filters (PF) (Doucet et al., 2001; Gordon et al., 1993; Kitagawa, 1996; Van Leeuwen, 2009) are an attractive family of nonlinear and non-Gaussian methods. This family of filters is known to suffer from filtering degeneracy, especially in

large-scale systems. Despite the fact that PFs do not force restrictive assumptions on the shape of the underlying probability distribution functions, they are not generally considered to be efficient without expensive tuning. While particle filtering algorithms have not yet been used operationally, their potential applicability for high dimensional problems is illustrated for example by (Rebeschini et al., 2015; Poterjoy, 2016; Llopis et al., 2017; Beskos et al., 2017; Potthast et al., 2017; Ades and van Leeuwen, 2015; Vetra-Carvalho et al., 2018). Another approach for non-Gaussian DA is to employ a Markov Chain Monte-Carlo (MCMC) algorithm, to directly sample the probability modes of the posterior distribution. This however, requires an accurate representation of the prior distribution, which is generally intractable in this context. Moreover, following a relaxed, e.g. Gaussian, prior assumption in nonlinear settings might be restrictive when a DA procedure is applied sequentially over more than one assimilation window. This is mainly due to fact that the prior distribution is a nonlinear transformation of the posterior of a previous assimilation cycle. Recently, an MCMC family of fully non-Gaussian DA algorithms that works by sampling the posterior were developed in (Attia and Sandu, 2015; Attia et al., 2015, 2016b, c, 2018; Attia, 2016). This family follows a Hamiltonian Monte-Carlo (HMC) approach for sampling the posterior, however, the HMC sampling scheme can be easily replaced with other algorithms suitable for sampling complicated, and potentially multimodal, probability distributions in high dimensional state spaces. Relaxing the Gaussian prior assumption is addressed in (Attia et al., 2018), where an accurate representation of the prior is constructed by fitting a Gaussian Mixture Model (GMM) to the forecast ensemble.

DATeS provides standard implementations of several flavors of the algorithms mentioned here. One can easily explore, test, or modify the provided implementations in DATeS, and add more methodologies. As discussed later, one can use existing components of DATeS, such as the implemented numerical models, or add new implementations to be used by other components of DATeS. However, it is worth mentioning that the initial version of DATeS (v1.0) is not meant to provide implementations of all state of the art DA algorithms; see e.g., (Vetra-Carvalho et al., 2018). DATeS however, provides an initial seed with example implementations, those could be discussed, and enhanced by the ever-growing community of DA researchers and experts. In the next Section, we provide a brief technical summary of the main components of DATeS v1.0.

## 3   DATeS Implementation

DATeS seeks to capture, in an abstract form, the common elements shared by most DA applications and solution methodologies. For example, the majority of the ensemble filtering methodologies share nearly all the steps of the forecast phase, and a considerable portion of the analysis step. Moreover, all the DA applications involve common essential components such as linear algebra routines, model discretization schemes, and analysis algorithms.

Existing DA solvers have been implemented in different languages. For example, high-performance languages such as Fortran and C have been (and are still being) extensively used to develop numerically efficient model implementations, and linear algebra routines. Both Fortran and C allow for efficient parallelization because these two languages are supported by common libraries designed for distributed memory systems such as MPI, and shared memory libraries such as Pthreads and OpenMP. To make use of these available resources and implementations, one has to either rewrite all the different pieces in the same programming language, or have proper interfaces between the different new and existing implementations.

The philosophy behind the design of DATeS is that "*a unified DA testing suite has to be open-source, easy to learn, and able to reuse and extend available code with minimal effort*". Such a suite should allow for easy interfacing with external third-party code written in various languages, e.g., linear algebra routines written in Fortran, analysis routines written in Matlab, or "forecast" models written in C. This should help the researchers to focus their energy on implementing and testing their own analysis algorithms. The next section details several key aspects of the DATeS implementation.

## 3.1 DATeS architecture

The DATeS architecture abstracts, and provides a set of modules of, the four generic components of any DA system. These components are the linear algebra routines, a forecast computer model that includes the discretization of the physical processes, error models, and analysis methodologies. In what follows, we discuss each of these building blocks in more details, in the context of DATeS. We start with an abstract discussion of each of these components, followed by technical descriptions.

### 3.1.1 Linear algebra routines

The linear algebra routines are responsible for handling the data structures representing essential entities such as model state vectors, observation vectors, and covariance matrices. This includes manipulating an instance of the corresponding data. For example, a model state vector should provide methods for accessing/slicing and updating entries of the state vector, a method for adding two state vector instances, and methods for applying specific scalar operations on all entries of the state vector such as evaluating the square root or the logarithm.

### 3.1.2 Forecast model

The forecast computer model simulates a physical phenomena of interest such as the atmosphere, ocean dynamics, and volcanoes. This typically involves approximating the physical phenomena using a gridded computer model. The implementation should provide methods for creating and manipulating state vectors, and state-size matrices. The computer model should also provide methods for creating and manipulating observation vectors and observation-size matrices. The observation operator responsible for mapping state-size vectors into observation-size vectors should be part of the model implementation as well. Moreover, simulating the evolution of the computer model in time is carried out using numerical time integration schemes. The time integration scheme can be model-specific, and is usually written in a high-performance language for efficiency.

### 3.1.3 Error models

It is common in DA applications to assume a perfect forecast model, a case where the model is deterministic rather than stochastic. However, the background and observation errors need to be treated explicitly as they are essential in the formulation of nearly all DA methodologies. We refer to the DATeS entity responsible for managing and creating random vectors, sampled from a specific probability distribution function, as the "*error model*". For example a Gaussian error model would be completely set up by providing the first and second order moments of the probability distribution it represents.

### 3.1.4 Analysis algorithms

Analysis algorithms manipulate model states and observations by applying widely used mathematical operations to perform inference operations. The popular DA algorithms can be classified into filtering and smoothing categories. An assimilation algorithm, a filter or a smoother, is implemented to carry out a single DA cycle. For example, in the filtering framework, an assimilation cycle refers to assimilating data at a single observation time by applying a forecast and an analysis step. On the other hand, in the smoothing context, several observations available at discrete time instances within an assimilation window are processed simultaneously in order to update the model state at a given time over that window; a smoother is designed to carry out the assimilation procedure over a single assimilation window. For example, EnKF and 3D-Var fall in the former category, while EnKS and 4D-Var fall in the latter.

### 3.1.5 Assimilation experiments

In typical numerical experiments a DA solver is applied for several consecutive cycles to assess its long-term performance. We refer to the procedure of applying the solver to several assimilation cycles as the "assimilation process". The assimilation process involves carrying out the forecast and analysis cycles repeatedly, creating synthetic observations or retrieving real observations, updating the reference solution when available, and saving experimental results between consecutive assimilation cycle.

### 3.1.6 DATeS layout

The design of DATeS takes into account the distinction between these components, and separate them in design following an Object-Oriented Programming (OOP) approach. A general description of DATeS architecture is given in Figure 1.

The enumeration in Figure 1 (numbers from 1 to 4 in circles) indicates the order in which essential DATeS objects should be created. Specifically, one starts with an instance of a model. Once a model object is created, an assimilation object is instantiated, and the model object is passed to it. An assimilation process object is then instantiated, with a reference to the assimilation object passed to it. The assimilation process object iterates the consecutive assimilation cycles and save and/or output the results which can be optionally analyzed later using visualization modules.

All DATeS components are independent such as to maximize the flexibility in experimental design. However, each newly added component must comply to DATeS rules in order to guarantee interoperability with the other pieces in the package. DATeS provides base classes with definitions of the necessary methods. A new class added to DATeS, for example to implement a specific new model, has to inherit the appropriate model base class, and provide implementations of the inherited methods from that base class.

In order to maximize both flexibility and generalizability, we opted to handle configurations, inputs, and output of DATeS object, using "*configuration dictionaries*". Parameters passed to instantiate an object are passed to the class constructor in the form of key-value pairs in the dictionaries. See Section 4 for examples on how to properly configure and instantiate DATeS objects.

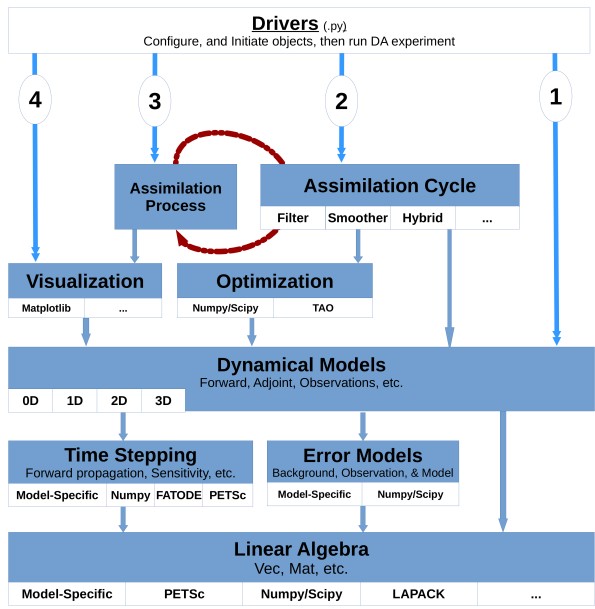

**Figure 1.** Diagram of the DATeS architecture.

## 3.2 Linear algebra classes

The main linear algebra data structures essential for almost all DA aspects are a) model state-size and observation-size vectors (also named state and observation vectors, respectively), and b) state-size and observation-size matrices (also named state and observation matrices, respectively). A state matrix is a square matrix of order equal to the model state space dimension. Similarly, an observation matrix is a square matrix of order equal to the model observation space dimension. DATeS makes a distinction between a state and observation linear algebra data structures. It is important to recall here that in large-scale applications, full state covariance matrices cannot be explicitly constructed in memory. Full state matrices should only be considered for relatively small problems, and for experimental purposes. In large-scale settings, where building state matrices is infeasible, low-rank approximations, or sparse representation of the covariance matrices could be incorporated. DATeS provides simple classes to construct sparse state and observation matrices for guidance.

Third-party linear algebra routines can have widely different interfaces and underlying data structures. For reusability, DATeS provides unified interfaces for accessing and manipulating these data structures using Python classes. The linear algebra classes are implemented in Python. The functionalities of the associated methods can be written either in Python, or in lower level languages using proper wrappers. A class for a linear algebra data structure enables updating, slicing, and manipulating an instance of the corresponding data structures. For example, a model state vector class provides methods that enable accessing/slicing and updating entries of the state vector, a method for adding two state vector instances, and methods for applying specific scalar operations on all entries of the state vector such as evaluating the square root or the logarithm. Once an instance

| Linear algebra base class | DATeS Implementation |
|---|---|
| state vector objects with access to all related vector operations | `state_vector_base.StateVectorBase` |
| observation vector objects with related vector operations | `observation_vector_base.ObservationVectorBase` |
| state matrix objects with methods implementing necessary matrix operations | `state_matrix_base.StateMatrixBase` |
| observation matrix objects providing methods for related matrix operations | `observation_matrix_base.ObservationMatrixBase` |

**Table 1.** DA filtering routines provided by the initial version of DATeS (v1.0)

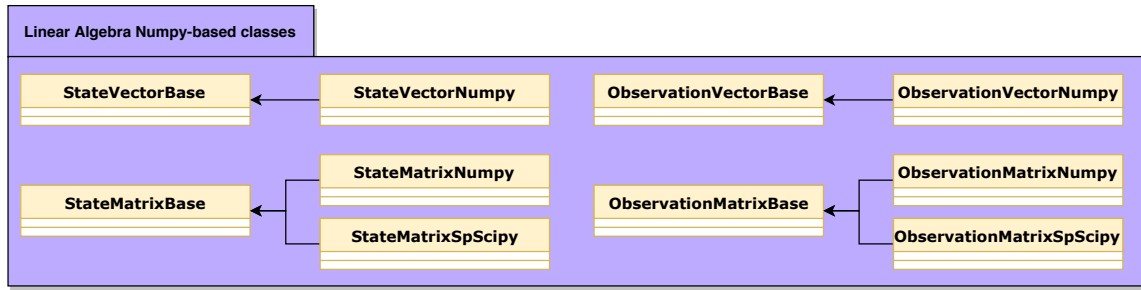

**Figure 2.** Python implementation of state vector, observation vector, state matrix, and observation matrix data structures. Both dense and sparse state and observation matrices are provided.

of a linear algebra data structure is created, all its associated methods are accessible via the standard Python dot operator. The linear algebra base classes provided in DATeS are summarized in Table 1.

Python special methods are provided in a linear algebra class to enable iterating a linear algebra data structure entries. Examples of these special methods include `__getitem__()`, `__setitem__()`, `__getslice__()`, `__setslice__()`,
etc. These operators make it feasible to standardize working with linear algebra data structures implemented in different languages or saved in memory in different forms.

DATeS provides linear algebra data structures represented as Numpy nd-arrays, and a set of Numpy-based classes to manipulate them. Moreover, Scipy-based implementation of sparse matrices is provided, and can be used efficiently in conjunction with both sparse and non-sparse data structures. These classes, shown in Figure 2, provide templates for designing more so-
phisticated extensions of the linear algebra classes.

### 3.3   Forecast model classes

Each numerical model needs an associated class providing methods to access its functionality. The unified forecast model class design in DATeS provides the essential tasks that can be carried out by the model implementation. Each model class in DATeS has to inherit the model base class: `models_base.ModelBase` or a class derived from it. A numerical model class
is required to provide access to the underlying linear algebra data structures and time integration routines. For example, each model class has to provide the method `state_vector()` that creates an instance of a state vector class, and the method `integrate_state()` that takes a state vector instance and time integration settings, and returns a trajectory (list of states)

| Forecast model | DATeS Implementation |
|---|---|
| the 3-variables Lorenz model (Lorenz, 1963) | `lorenz_models.Lorenz3` |
| Lorenz96 model (Lorenz, 1996) | `lorenz_models.Lorenz96` |
| Cartesian shallow-water equations model (Gustafsson, 1971; Navon and De-Villiers, 1986) | `cartesian_swe_mode.CartesianSWE` |
| Quasi-geostrophic (QG) model with double-gyre wind forcing and bi-harmonic friction (Sakov and Oke, 2008) written in Fortran, with a F2Py wrapper. | `qg_1p5_model.QG1p5` |

**Table 2.** DA filtering routines provided by the initial version of DATeS (v1.0)

evaluated at specified future times. The base class provided in DATeS contains definitions of all the methods that need to be supported by a numerical model class. The package DATeS v1.0 includes implementations of several popular test models summarized in Table 2.

While some linear algebra and the time integration routines are model-specific, DATeS also implements general-purpose linear algebra classes and time integration routines that can be reused by newly created models. For example, the general integration class `FatODE_ERK_FWD` is based on FATODE (Zhang and Sandu, 2014) explicit Runge-Kutta (ERK) forward propagation schemes.

### 3.4    Error models classes

In many DA applications the errors are additive, and are modeled by random variables normally distributed with zero mean
and a given or an unknown covariance matrices. DATeS implements Numpy-based functionality for background, observation, and model errors as guidelines for more sophisticated problem-dependent error models. The Numpy-based error models in DATeS are implemented in the module `error_models_numpy`. These classes are derived from the base class `ErrorsModelBase` and provide methodologies to sample the underlying probability distribution, evaluate the value of the density function, and generate statistics of the error variables based on model trajectories and the settings of the error model.
Note that, while DATeS provides implementations for Gaussian error models, the Gaussian assumption itself is not restrictive. Following the same structure, or by inheritance, one can easily create Non-Gaussian error models with minimal efforts. Moreover, the Gaussian error models provided by DATeS support both correlated and uncorrelated errors, and it constructs the covariance matrices accordingly. The covariance matrices are stored in appropriate sparse formats, unless a dense matrix is explicitly requested. Since these covariance matrices are either state or observation matrices, they provide access to all proper
linear algebra routines. This means that, the code written with access to an observation error model, and its components should work for both correlated and uncorrelated observations.

### 3.5    Assimilation classes

Assimilation classes are responsible for carrying out a single assimilation cycle (i.e., over one assimilation window) and optionally printing or writing the results to files. For example, an EnKF object should be designed to carry out one cy-

| Filtering Algorithm | DATeS Implementation |
|---|---|
| standard Kalman filter equations (Kalman and Bucy, 1961; Kalman, 1960) | `KF.KalmanFilter` |
| perturbed-observation (stochastic) EnKF (Burgers et al., 1998; Houtekamer and Mitchell, 1998) | `EnKF.EnKF` |
| deterministic EnKF (Sakov and Oke, 2008) | `EnKF.DEnKF` |
| Ensemble transform Kalman filter (ETKF) (Bishop et al., 2001) | `EnKF.ETKF` |
| local least squares EnKF (Anderson, 2003) | `EnKF.LLSEnKF` |
| Hybrid Monte-Carlo (HMC) sampling filter (Attia and Sandu, 2015) | `hmc_filter.HMCFilter` |
| Family of cluster sampling filters (Attia et al., 2018) | `multi_chain_mcmc_filter.MultiChainMCMC` |
| A vanilla implementation of the particle filter (Gordon et al., 1993) | `PF.PF` |

**Table 3.** DA filtering routines provided by the initial version of DATeS v1.0

cle consisting of the "forecast" and the "analysis" steps. The basic assimilation objects in DATeS are a filtering object, a smoothing object, and a hybrid object. DATeS provides the common functionalities for filtering objects in the base class `filters_base.FiltersBase`; all derived filtering classes should have it as a super class. Similarly, smoothing objects are to be derived from the base class `smoothers_base.SmoothersBase`. A hybrid object can inherit methods from both filtering and smoothing base classes.

A model object is passed to the assimilation object constructor via configuration dictionaries to give the assimilation object access to the model-based data structures and functionalities. The settings of the assimilation object, such as the observation time, the assimilation time, the observation vector, and the forecast state or ensemble, are also passed to the constructor upon instantiation, and can be updated during runtime.

Table 3 summarizes the filters implemented in the initial version of the package, that is DATeS v1.0. Each of these filtering classes can be instantiated and run with any of the DATeS model objects. Moreover, DATeS provides simplified implementations of both 3D-Var, and 4D-var assimilation schemes. The objective function, e.g. the negative log-posterior, and the associated gradient are implemented inside the smoother class, and require the tangent linear model to be implemented in the passed forecast model class. The adjoint is evaluated using FATODE following a checkpointing approach, and the optimization step is carried out using Scipy optimization functions. The settings of the optimizer can be fine-tuned via the configurations dictionaries. The 3D- and 4D-Var implementations provided by DATeS are experimental, and are provided as a proof of concept. The variational aspects of DATeS are being continuously developed and will be made available in future releases of the package.

Covariance inflation and localization are ubiquitously used in all ensemble-based assimilation systems. These two methods are used to counteract the effect of using ensembles of finite size. Specifically, covariance inflation counteracts the loss of variance incurred in the analysis step, and works by inflating the ensemble members around their mean. This is carried out by magnifying the spread of ensemble members around their mean, by a predefined inflation factor. The inflation factor could be a scalar, i.e. space-time independent, or even varied over space and/or time. Localization, on the other hand, mitigates the accumulation of long-range spurious correlations. Distance-based covariance localization is widely used in geoscientific sciences, and applications, where correlations are damped out with increasing distance between grid points. The performance of the assimilation algorithm is critically dependent on tuning the parameters of these techniques. DATeS provide basic utility

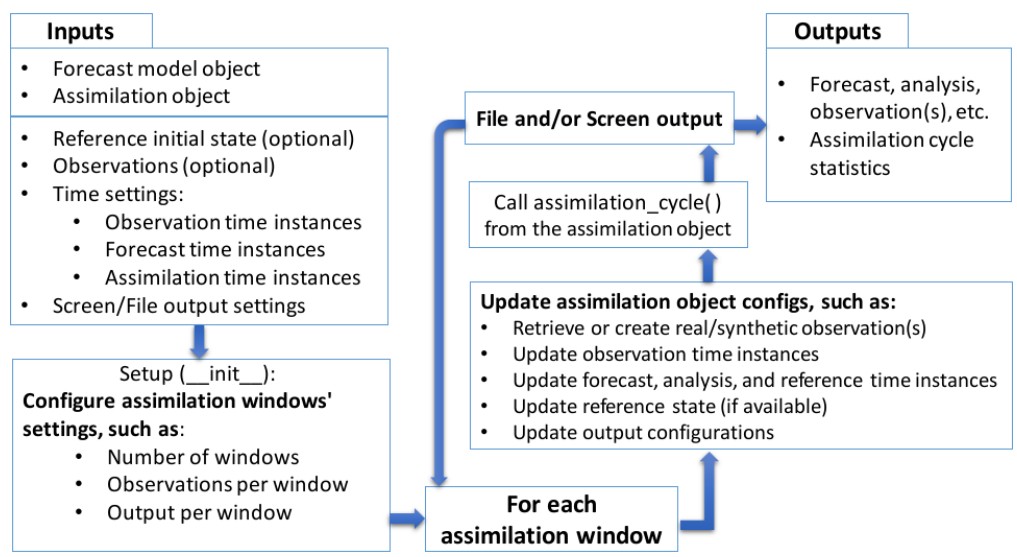

**Figure 3.** The assimilation process in DATeS.

functions (see Section 3.7) for carrying out inflation and localization those can be used in different forms based on the specific implementation of the assimilation algorithms. The work in (Attia and Constantinescu, 2018) reviews inflation and localization and presents a framework for adaptive tuning of the parameters of these techniques, with all implementations and numerical experiments carried out entirely in DATeS.

## 3.6  Assimilation process classes

A common practice in sequential DA experimental settings, is to repeat an assimilation cycle over a given timespan, with similar or different settings at each assimilation window. For example, one may repeat a DA cycle on several time intervals with different output settings; e.g. to save and print results only every fixed number of iterations. Alternatively, the DA process can be repeated over the same time interval with different assimilation settings to test and compare results. We refer to this procedure as an "assimilation process". Examples of numerical comparisons, carried out used DATeS, can be found in (Attia et al., 2018; Attia and Constantinescu, 2018), and in Section 4.6.

`assimilation_process_base.AssimilationProcess` is the base class from which all assimilation process objects are derived. When instantiating an assimilation process object, the assimilation object, the observations and the assimilation time instances, are passed to the constructor through configuration dictionaries. As a result, the assimilation process object has access to the model and its associated data structures and functionalities through the assimilation object.

The assimilation process object either retrieves real observations, or creates synthetic observations at the specified time instances of the experiment. Figure 3 summarizes DATeS assimilation process functionality.

| Module name | Functionality provided |
|---|---|
| `_utility_configs` | handles configuration dictionaries, including aggregating, reading, and writing configuration dictionaries |
| `_utility_stat` | evaluate statistical quantities such as moments of an ensemble (e.g. list of model state or observation objects) |
| `_utility_machine_learning` | carry out machine learning algorithms such as fitting a Gaussian Mixture Model to an ensemble |
| `_utility_data_assimilation` | carry out general DA tasks such as ensemble inflation, covariance localization, and evaluating performance metrics including root-mean-squared errors (RMSE), and rank histogram uniformity measures. |

**Table 4.** A sample of the modules wrapped by the main utility module `dates_utility`.

## 3.7 Utility modules

Utility modules provide additional functionality, such as `_utility_configs` module which provides functions for reading, writing, validating, and aggregating configuration dictionaries. In DA, an ensemble is a collection of state or observation vectors. Ensembles are represented in DATeS as lists of either state, or observation vector objects. The utility modules include functions responsible for iterating over ensembles to evaluate ensemble-related quantities of interest, such as ensemble mean, ensemble variance/covariance, and covariance trace. Covariance inflation and localization are critically important for nearly all ensemble-based assimilation algorithms. DATeS abstracts tools and functions common to assimilation methods, such as inflation and localization where they can be easily imported and reused by newly developed assimilation routines. The utility module in DATeS provide methods to carry out these procedures in various modes, including state space and observation space localization. Moreover, DATeS supports space-dependent covariance localization, i.e. it allows varying the localization radii and inflation factors over both space and time.

Ensemble-based assimilation algorithms often require matrix representation of ensembles of model states. In DATeS, ensembles are represented as lists of states, rather than full matrices of size $N_{state} \times N_{ens}$. However, it provides utility functions capable of efficiently calculating ensemble statistics, including ensemble variances, and covariance trace. Moreover, DATeS provides matrix-free implementations of the operations that require ensembles of states, such as a matrix-vector product, where the matrix is involved is a representation of an ensemble of states.

The module `dates_utility` provides access to all utility functions in DATeS. In fact, this module wraps the functionality provided by several other specialized utility routines, including the sample given in Table 4. The utility module provides other general functions such as to handling file downloading, and functions for file I/O. For a list of all functions in the utility module, see the User's Manual (Attia et al., 2016a).

## 4 Using DATeS

The sequence of steps needed to run a DA experiment in DATeS is summarized in Figure 4. This section is devoted to explaining these steps in the context of a working example that uses the QG-1.5 model (Sakov and Oke, 2008) and carries out DA using a standard EnKF formulation.

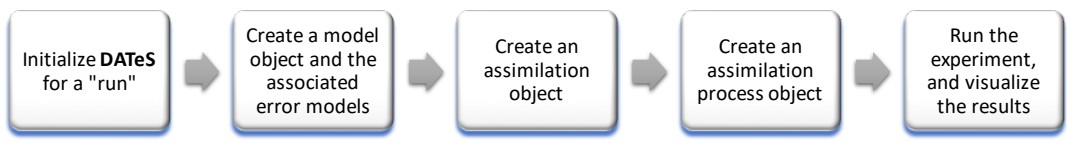

**Figure 4.** The sequence of essential steps required in order to run a DA experiment in DATeS.

```
import dates_setup
dates_setup.initialize_dates()
```

**Figure 5.** Initialize the DATeS run.

## 4.1 Step1: initialize DATeS

Initializing a DATeS run involves defining the root directory of DATeS as an environment variable, and adding the paths of DATeS source modules to the system path. This can be done by executing code Snippet in Figure 5 in DATeS root directory.

## 4.2 Step2: create a model object

QG-1.5 is a nonlinear 1.5-layer reduced-gravity QG model with double-gyre wind forcing and bi-harmonic friction (Sakov and Oke, 2008).

### 4.2.1 Quasi-geostrophic model

This model is a numerical approximation of the equations:

$$q_t = \psi_x - \varepsilon J(\psi, q) - A\Delta^3 \psi + 2\pi \sin(2\pi y),$$
$$q = \Delta\psi - F\psi, \tag{9}$$
$$J(\psi, q) \equiv \psi_x q_x - \psi_y q_y,$$

where $\Delta := \partial^2/\partial x^2 + \partial^2/\partial y^2$ and $\psi$ is the surface elevation. The values of the model coefficients in (9) are obtained from (Sakov and Oke, 2008), and are described as follows: $F = 1600$, $\varepsilon = 10^{-5}$, and $A = 2 \times 10^{-12}$. The domain of the model is a $1 \times 1$ [space units] square, with $0 \leq x \leq 1$, $0 \leq y \leq 1$, and is discretized by a grid of size $129 \times 129$ (including boundaries). The boundary conditions used are $\psi = \Delta\psi = \Delta^2\psi = 0$. The dimension of the model state vector is $N_{\text{state}} = 16641$. This is a synthetic model where the scales are not relevant, and we use generic space, time, and solution amplitude units. The

time integration scheme used is the fourth-order Runge-Kutta scheme with a time step $1.25$ [time units]. The model forward propagation core is implemented in Fortran. The QG-1.5 model is run over $1000$ model time steps, with observations made available every $10$ time steps.

```
from qg_1p5_model import QG1p5
model = QG1p5(model_configs = dict(MREFIN=7, observation_operator_type='linear',
                                    observation_vector_size=300, observation_error_variances=4.0))
```

**Figure 6.** Create the QG model object.

```
# create an initial ensemble
ens_size = 20
initial_ensemble = model.create_initial_ensemble(ensemble_size=ens_size, ensemble_from_repo=True)

# create filter object
from EnKF import DEnKF
denkf_filter_configs = dict(model=model,
                            analysis_ensemble=initial_ensemble,
                            ensemble_size=ens_size,
                            inflation_factor=1.06,
                            localize_covariances=True,
                            localization_method='covariance_filtering',
                            localization_radius=12,
                            localization_function='Gaspari−Cohn')
denkf_filter = DEnKF(filter_configs=denkf_filter_configs, output_configs=dict(file_output_moment_only=False))
```

**Figure 7.** Create a DEnKF filtering object.

### 4.2.2 Observations and observation operators

We use a standard linear operator to observe $300$ components of $\psi$ with observation error variance set to $4.0$ [units squared]. The observed components are uniformly distributed over the state vector length, with an offset that is randomized at each filtering cycle. Synthetic observations are obtained by adding white noise to measurements of the see height level (SSH) extracted from a reference model run with lower viscosity. To create a QG model object with these specifications, one executes code Snippet in Figure 6.

### 4.3 Step3: create an assimilation object

One now proceeds to create an assimilation object. We consider a deterministic implementation of EnKF (DEnKF) with ensemble size equal to 20, and parameters tuned optimally as suggested in (Sakov and Oke, 2008). Covariance localization is applied via a Hadamard product (Houtekamer and Mitchell, 2001). The localization function is Gaspari-Cohn (Gaspari and Cohn, 1999) with a localization radius of 12 grid cells. The localization is carried out in the observation space by decorrelating both $\mathbf{HB}$ and $\mathbf{HBH}^T$, where $\mathbf{B}$ is the ensemble covariance matrix, and $\mathbf{H}$ is the linearized observation operator. In the present setup, the observation operator $\mathcal{H}$ is linear, and thus $\mathbf{H} = \mathcal{H}$.

Ensemble inflation is applied to the analysis ensemble of anomalies at the end of each assimilation cycle of DEnKF with an inflation factor of $1.06$. Code Snippet in Figure 7 creates a DEnKF filtering object with these settings.

```
# create observation and assimilation checkpoints
import numpy as np
da_checkpoints = obs_checkpoints = np.arange(0, 1250.001, 12.5)

# create sequential filtering_process object
from filtering_process import FilteringProcess
ref_IC = model._reference_initial_condition.copy()
experiment = FilteringProcess(assimilation_configs=dict(filter=denkf_filter,
                                                        da_checkpoints=da_checkpoints,
                                                        ref_initial_condition=ref_IC,
                                                        obs_checkpoints=obs_checkpoints),
                              output_configs = dict(scr_output=True, scr_output_iter=1,
                                                    file_output=True, file_output_iter=1))
```

**Figure 8.** Create a filter process object to carry out DEnKF filtering using the QG model.

```
experiment.recursive_assimilation_process()
```

**Figure 9.** Run the filtering experiment.

Most of the methods associated to the DEnKF object will raise exceptions if immediately invoked at this point. This is because several keys in the filter configuration dictionary such as the observation, the forecast time, the analysis time, and the assimilation time, are not yet appropriately assigned. DATeS allows creating assimilation objects without these options to maximize flexibility. A convenient approach is to create an assimilation process object that, among other tasks, can properly

update the filter configurations between consecutive assimilation cycles.

### 4.4 Step4: create an assimilation process

We now test DEnKF with QG model by repeating the assimilation cycle over a timespan from $0$ to $1250$ with offsets of $12.5$ time units between each two consecutive observation/assimilation time. An initial ensemble is created by the numerical model object. An experimental timespan is set for observations and assimilation. Here, the assimilation time instances

`da_checkpoints` are the same as the observation time instances `obs_checkpoints`, but they can in general be different, leading to either synchronous or asynchronous assimilation settings. This is implemented in the code Snippet in Figure 8. Here `experiment` is responsible for creating synthetic observations at all time instances defined by `obs_checkpoints` (except the initial time). To create synthetic observations the truth at the initial time (0 in this case) is obtained from the model and is passed to the filtering process object `experiment`, which in turn propagates it forward in time to assimilation time

points.

Finally, the assimilation experiment is executed by running code Snippet in Figure 9.

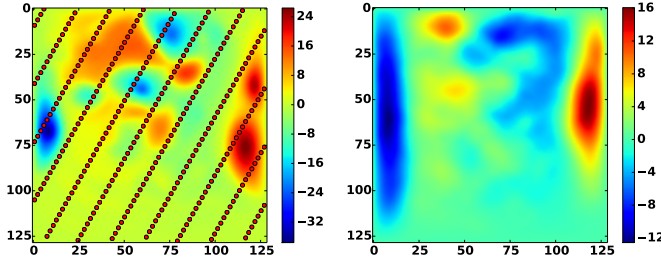

**Figure 10.** The QG-1.5 model. The truth (reference state) at the initial time (t=0) of the assimilation experiment is shown in the left panel. The red dots indicate the locations of observations for one of the test cases employed. The initial forecast state, taken as the average of the initial ensemble at time t=0, is shown in the right panel.

## 4.5 Experiment results

The filtering results are printed to screen and are saved to files at the end of each assimilation cycle as instructed by the `output_configs` dictionary of the object `experiment`. The output directory structure is controlled via the options in the output configurations dictionary `output_configs` of the `FilteringProcess` object, i.e. `experiment`. All re-
sults are saved in appropriate sub-directories under a main folder named `Results` in the root directory of DATeS. We will refer to this directory henceforth as DATeS results directory. The default behavior of a `FilteringProcess` object is to create a folder named `Filtering_Results` in DATeS results directory, and to instruct the filter object to save/output the results every `file_output_iter` whenever the flag `file_output` is turned on. Specifically, the `DEnKF` object creates three directories named `Filter_Statistics`, `Model_States_Repository`, and `Observations_Repository`
respectively. The root mean-squared (RMS) forecast and analysis errors are evaluated at each assimilation cycle, and are written to a file under `Filter_Statistics` directory. The output configurations of the filter object of the `DEnKF` class, i.e. `denkf_filter`, instructs the filter to save all ensemble members (both forecast and analysis) to files by setting the value of the option `file_output_moment_only` to `False`. The true solution (reference state), the analysis ensemble, and the forecast ensembles are all saved under the directory `Model_States_Repository`, while the observations are saved
under the directory `Observations_Repository`. We note that while here we illustrate the default behavior, the output directories are fully configurable.

Figure 10 shows the reference initial state of the QG model, an example of the observational grid used, and an initial forecast state. The initial forecast state in Figure 10 is the average of an initial ensemble collected from a long run of the QG model.

The true field, the forecast errors, and the DEnKF analyses errors at different time instances are shown in Figure 11.
Typical solution quality metrics in the ensemble-based DA literature include RMSE plots and Rank (Talagrand) histograms (Anderson, 1996; Candille and Talagrand, 2005).

Upon termination of a DATeS run, executable files can be cleaned up by calling the function `clean_executable_files()` available in the utility module (see code Snippet in Figure 12).

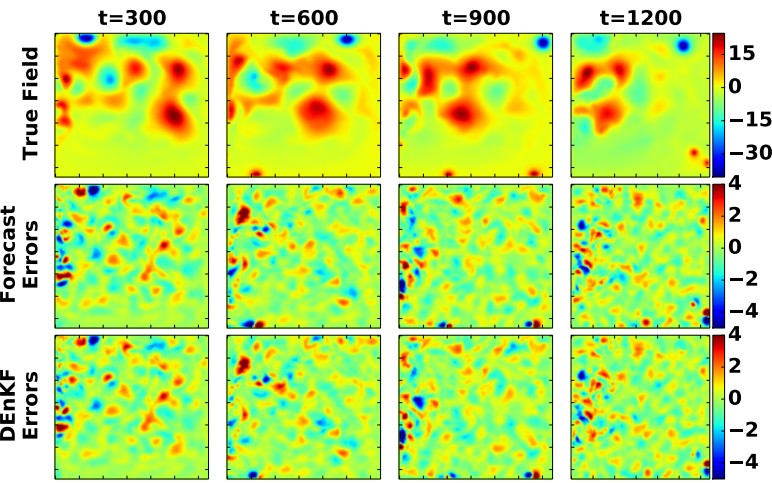

**Figure 11.** Data assimilation results. The reference field $\psi$, the forecast errors, and the analysis errors at $t = 300$, $t = 600$, $t = 900$, $t = 1200$ [time units]. Here the forecast error is defined as the reference field minus the average of the forecast ensemble, and the analysis error is the reference field minus the average of the analysis ensemble.

```
# cleanup executables and temporary modules
import dates_utility as utility
utility.clean_executable_files()
```

**Figure 12.** Cleanup DATeS executable files.

## 4.6   DATeS for benchmarking

### 4.6.1   Performance metrics

In the linear settings, the performance of an ensemble-based DA filter could be judged based on two factors. Firstly, convergence explained by its ability to track the truth, and secondly by the quality of the flow-dependent covariance matrix generated given the analysis ensemble.

The convergence of the filter is monitored by inspecting the root mean squared error (RMSE), which represents an ensemble-based standard deviation of the difference between reality, or truth, and the model-based prediction. In synthetic experiments, where the model representation of the truth is known, the RMSE reads

$$\mathbf{RMSE} = \sqrt{\frac{1}{\mathrm{N_{state}}} \sum_{i=1}^{\mathrm{N_{state}}} (x_i - x_i^{\mathrm{True}})^2}, \tag{10}$$

where $\mathbf{x} = (x_1, x_2, \ldots, x_{\mathrm{N_{state}}})^T \in \mathbb{R}^{\mathrm{N_{state}}}$ is the prediction at a given time instant, e.g., the forecast ensemble mean, and $\mathbf{x}^{\mathrm{True}} \in \mathbb{R}^{\mathrm{N_{state}}}$ is the verification, e.g., the true model state at the same time instant. For real applications, the states are

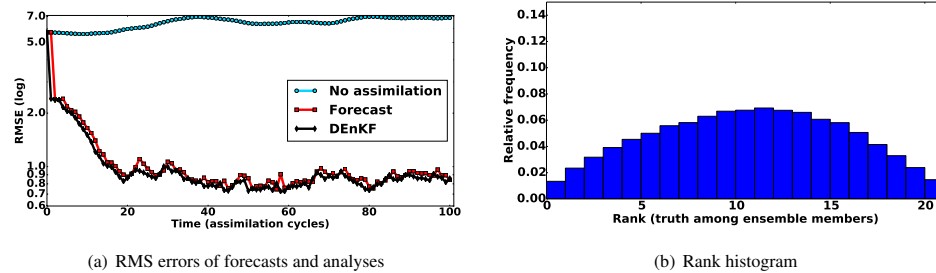

(a) RMS errors of forecasts and analyses        (b) Rank histogram

**Figure 13.** Data assimilation results. In panel (a) "no assimilation" refers to the RMSE of the initial forecast (the average of the initial forecast ensemble) propagated forward in time over the 100 cycles without assimilating observations into it. The rank histogram of where the truth ranks among analysis ensemble members is shown in panel (b). The ranks are evaluated for every $13^{th}$ variable in the state vector (past the correlation bound) after 100 assimilation cycles.

generally replaced with observations. The rank (Talagrand) histogram (Anderson, 1996; Candille and Talagrand, 2005), could be used to assess the spread of the ensemble, and its coverage to the truth. Generally speaking, the rank histogram plots the rank of the truth (or observations) compared to the ensemble members (or equivalent observations), ordered increasingly in magnitude. A nearly-uniform rank histogram is desirable, and suggests that the truth is indistinguishable from the ensemble members. A mound rank histogram indicates overdispersed ensemble, a while U-shaped histogram indicates underdispersion. However, mound rank histograms are rarely seen in practice, especially for large-scale problems. See e.g., (Hamill, 2001) for a mathematical description and a detailed discussion on the usefulness and interpretation of rank histograms.

Figure 13 (a) shows an RMSE plot of the results of the experiment presented in 4.5. The histogram of the rank statistics of the truth, compared to the analysis ensemble, is shown in Figure 13 (b).

For benchmarking, one needs to generate scalar representations of the RMSE and the uniformity of a rank histogram, of a numerical experiment. The average RMSE can be used to compare the accuracy of a group of filters. To generate a scalar representation of the uniformity of a rank histogram, we fit a Beta distribution to the rank histogram, scaled to the interval $[0, 1]$, and evaluate the Kullback-Leibler (KL) divergence (Kullback and Leibler, 1951) between the fitted distribution, and a uniform distribution[1]. We consider a small, e.g. closer to 0, KL distance to be an indication of a nearly-uniform rank histogram, and consequently an indication of a well-dispersed ensemble. An alternative measure of rank histogram uniformity, is to average the absolute distances of bins' heights from a uniformly distributed rank histogram (Bessac et al., 2018). DATeS provides several utility functions to calculate such metrics for a numerical experiment.

Figure 14 shows several rank histograms, along with uniform distribution, and fitted Beta distributions. The KL-divergence measure is indicated under each panel. Results in Figure 14, suggest that the fitted Beta distribution parameters give, in most cases, a good scalar description of the shape of the histogram. Moreover, one can infer the shape of the rank histogram from the parameters of the fitted Beta distribution. For example, if $\alpha > 1$, and $\beta > 1$, the histogram has a mound shape, and is U-shaped if $\alpha < 1$, and $\beta < 1$. The histogram is closer to uniformity as the parameters $\alpha, \beta$ both approach 1. Table 5 shows both KL

---

[1] The KL divergence between two Beta distributions $\text{Beta}(\alpha, \beta)$, and $\text{Beta}(\alpha', \beta')$ is

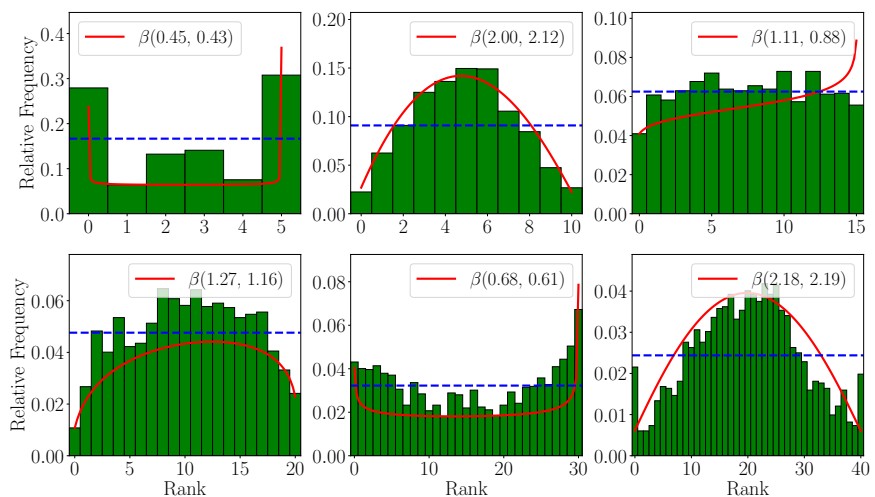

**Figure 14.** Rank histograms, with fitted Beta distributions. The KL-divergence measure is indicated under each panel.

**Table 5.** Measures of uniformity of the rank histograms shown in Figure 14.

| Panel | 1 | 2 | 3 | 4 | 5 | 6 |
|---|---|---|---|---|---|---|
| $D_{\mathrm{KL}}(\beta|\mathcal{U})$ | 0.198 | 0.231 | 0.022 | 0.018 | 0.065 | 0.272 |
| Average distance to $\mathcal{U}$ | 0.085 | 0.038 | 0.005 | 0.011 | 0.008 | 0.010 |

distance between fitted Beta distribution with respect to a uniform, and the average distances between histogram bins and a uniform.

### 4.6.2 Benchmarking

The architecture of DATeS makes is easy to generate benchmarks for a new experiment. For example, one can write short
5    scripts to iterate over a combination of settings of a filter to find the best possible results. As an example, consider the standard 40-variables Lorenz-96 model (Lorenz, 1996) described by the equations

$$\frac{dx_i}{dt} = x_{i-1}\left(x_{i+1} - x_{i-2}\right) - x_i + F\,;\, i = 1, 2, \ldots, 40\,, \tag{11}$$

where $\mathbf{x} = (x_1, x_2, \ldots, x_{40})^T \in \mathbb{R}^{40}$ is the state vector, with periodic boundaries, i.e. $x_0 \equiv x_{40}$, and the forcing parameter is set to $F = 8$. These settings make the system chaotic (Lorenz and Emanuel, 1998) and are widely used in synthetic settings
10    for geoscientific applications. Adjusting the inflation factor, and the localization radius for EnKF filter is crucial. Consider the

---

$$D_{\mathrm{KL}}(\mathrm{Beta}(\alpha, \beta)\,|\,\mathrm{Beta}(\alpha'\,\beta')) = \ln\Gamma(\alpha + \beta) - \ln(\alpha\beta) - \ln\Gamma(\alpha' + \beta') + \ln(\alpha'\,\beta') + (\alpha - \alpha')\left(\psi(\alpha) - \psi(\alpha')\right) + (\beta - \beta')\left(\psi(\beta) - \psi(\beta')\right)$$

where $\psi(\cdot) = \frac{\Gamma'(\cdot)}{\Gamma(\cdot)}$ is the digamma function, i.e. the logarithmic derivative of the gamma function. Here, we set $Beta(\alpha', \beta')$ to a uniform distribution by setting $\alpha' = \beta' = 1$.

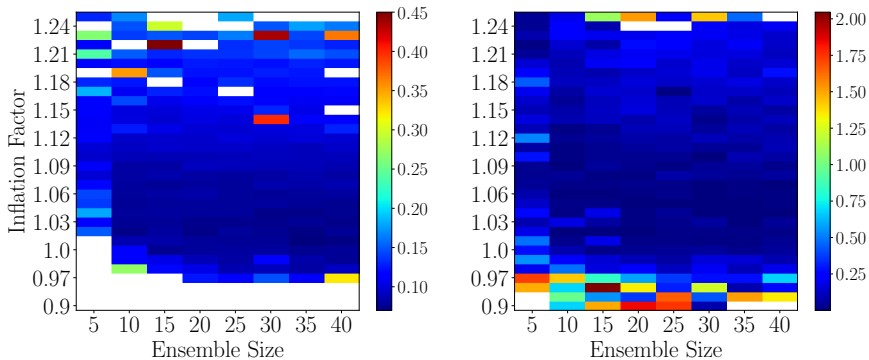

**Figure 15.** Data assimilation results with DEnKF applied to Lorenz-96 system. RMSE results on, a log-scale, are shown in the first panel. The KL-distances between the analysis rank histogram and a uniform rank histogram are shown in the second panel. The localization radius is fixed to 4.

case where one is testing an adaptive inflation scheme, and would like to decide on the ensemble size, and the benchmark inflation factor to be used. As an example of benchmarking, we run the following experiment, over a time interval $[0,30]$ [units], where (11) is integrated forward in time using a fourth-order Runge-Kutta scheme with model step size $0.005$ [units]. Assume that synthetic observations are generated every 20 model steps, where every other entry of the model state is observed.

We test the DEnKF algorithm, with the fifth-order piecewise-rational function of Gaspari and Cohn (Gaspari and Cohn, 1999) for covariance localization. The localization radius is held constant, and is set to $l = 4$, while the inflation factor is varied for each experiment. The experiments are repeated for ensemble sizes $N_{ens} = 5, 10, \ldots, 40$. We report the results over the second two-thirds of the experiments timespan, i.e. over the interval $[10, 30]$ to avoid spinup artifacts. This interval consisting of the last 200 assimilation cycles out of 300, will be referred to as the "testing timespan". Any experiment that results in an average

RMSE of more than $0.65$ over the testing timespan, is discarded, and the filter used is seen to diverge. The numerical results are summarized in Figures 15, and 16.

Figure 15 shows the average RMSE results, and the KL-distances between a Beta distribution fitted to the analysis rank histogram of each experiment, and a uniform distribution. These plots, give a preliminary idea of the plausible regimes of both ensemble size, and inflation factor that should be used to achieve the best performance of the filter used, under the

15 current experimental settings. For example, for an ensemble size $N_{ens} = 20$, the inflation factor should be set approximately to $1.01 - 1.07$ to give both a small RMSE and an analysis rank histogram close to uniform.

Concluding the best inflation factor, for a given ensemble size, based on Figure 15, however could be tricky. Figure 16 shows the inflation factors resulting in minimum average RMSE and minimum KL distance to uniformity. Specifically, for each ensemble size a red triangle refers to the experiment that resulted in minimum average RMSE over the testing timespan,

out of all benchmarking experiments carried out with this ensemble size. Similarly, the experiment that yielded minimum KL-divergence to a uniform rank histogram, is indicated by a blue tripod.

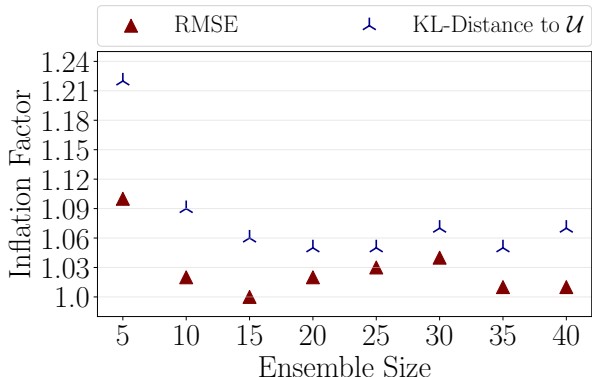

**Figure 16.** Data assimilation results with DEnKF applied to Lorenz-96 system. The minimum average RMSE over the interval $[10, 30]$ is indicated by red triangles. The minimum average KL distance between the analysis rank histogram and a uniformly distributed rank histogram $[10, 30]$ is indicated by blue tripods. We show the results for every choice of the ensemble size. The localization radius is fixed to 4.

To answer the question about the ensemble size, we pick the ensemble size $N_{ens} = 25$, given the current experimental setup. The reason is that $N_{ens} = 25$ is the smallest ensemble size that yields small RMSE, and well-dispersed ensemble as explained by Figure 16. As for the benchmark inflation factor, the results in Figures 16 show that for an ensemble size $N_{ens} = 25$, the best choice of an inflation factor is approximately $1.03 - 1.05$, for Gaspari-Cohn localization with a fixed radius of 4.

Despite being a relatively easy process, unfortunately generating a set of benchmarks for all possible combinations of numerical experiments is a time-consuming process, and is better carried out by the DA community. Some example scripts for generating and plotting benchmarking results are included in the package for guidance.

Note that, when the Gaussian assumption is severely violated, standard benchmarking tools, such as RMSE and rank histograms, should be replaced with, or at least supported by, tools capable of assessing ensemble coverage of the posterior

distribution. In such cases, MCMC methods, including those implemented in DATeS (Attia and Sandu, 2015; Attia et al., 2018; Attia, 2016), could be used as a benchmarking tool (Law and Stuart, 2012).

## 5    Extending DATeS

DATeS aims at being a collaborative environment, and is designed such that adding DA components to the package is as easy and flexible as possible. This section describes how new implementations of components such as numerical models and

assimilation methodologies can be added to DATeS.

The most direct approach is to write the new implementation completely in Python. This, however, may sacrifice efficiency, or may not be feasible when existing code in other languages needs to be reused. One of the main characteristics of DATeS is the possibility of incorporating code written in low level languages. There are several strategies that can be followed to

interface existing C or Fortran code with DATeS. Amongst the most popular tools are SWIG, and F2Py for interfacing Python code with existing implementations written in C and Fortran, respectively.

Whether the new contribution is written in Python, in C, or in Fortran,an appropriate Python class that inherits the corresponding base class, or a class derived from it, has to be created. The goal is to design new classes those are conformable with the existing structure of DATeS and can interact appropriately with new as well as existing components.

## 5.1 Adding a numerical model class

A new model class has to be created as a subclass of `ModelsBase`, or a class derived from it. The base class `ModelsBase`, similar to all base classes in DATeS, contains headers of all the functions that need to be provided by a model class to guarantee that it interacts properly with other components in DATeS.

The first step is to grant the model object access to linear algebra data structures, and to error models. Appropriate classes should be imported in a numerical model class:

- Linear algebra: state vector, state matrix, observation vector, and observation matrix, and
- Error models: background, model, and observation error models.

This gives the model object access to model-based data structures, and error entities necessary for DA applications. Figure 17 illustrates a class of a numerical model named "MyModel", along with all the essential classes imported by it.

The next step is to create Python-based implementations for the model functionalities. As shown in Figure 17, the corresponding methods have descriptive names in order to ease the use of DATeS functionality. For example, the method `state_vector()` creates (or initializes) a state vector data structure. Details of each of the methods in Figure 17 are given in the DATeS User Manual (Attia et al., 2016a).

As an example, suppose we want to create a model class name `MyModel` using Numpy and Scipy (for sparse matrices) linear algebra data structures. Code Snippet in Figure 18 shows the implementation of such class.

Note that in order to guarantee extensibility of the package we have to fix the naming of the methods associated with linear algebra classes, and even if only binary files are provided, the Python-based linear algebra methods must be implemented. If the model functionality is fully written in Python, the implementation of the methods associated with a model class is straightforward, as illustrated in (Attia et al., 2016a). On the other hand, if a low level implementation of a numerical model is given, these methods wrap the corresponding low level implementation.

## 5.2 Adding an assimilation class

The process of adding a new class for an assimilation methodology is similar to creating a class for a numerical model, however it is expected to require less effort. For example, a class implementation of a filtering algorithm uses components and tools provided by the passed model, and by the encapsulated linear algebra data structures and methods. Moreover, filtering algorithms belonging to the same family, such as different flavors of the well-known EnKF, are expected to share a considerable amount of infrastructure. Python inheritance enables the reuse of methods and variables from parent classes.

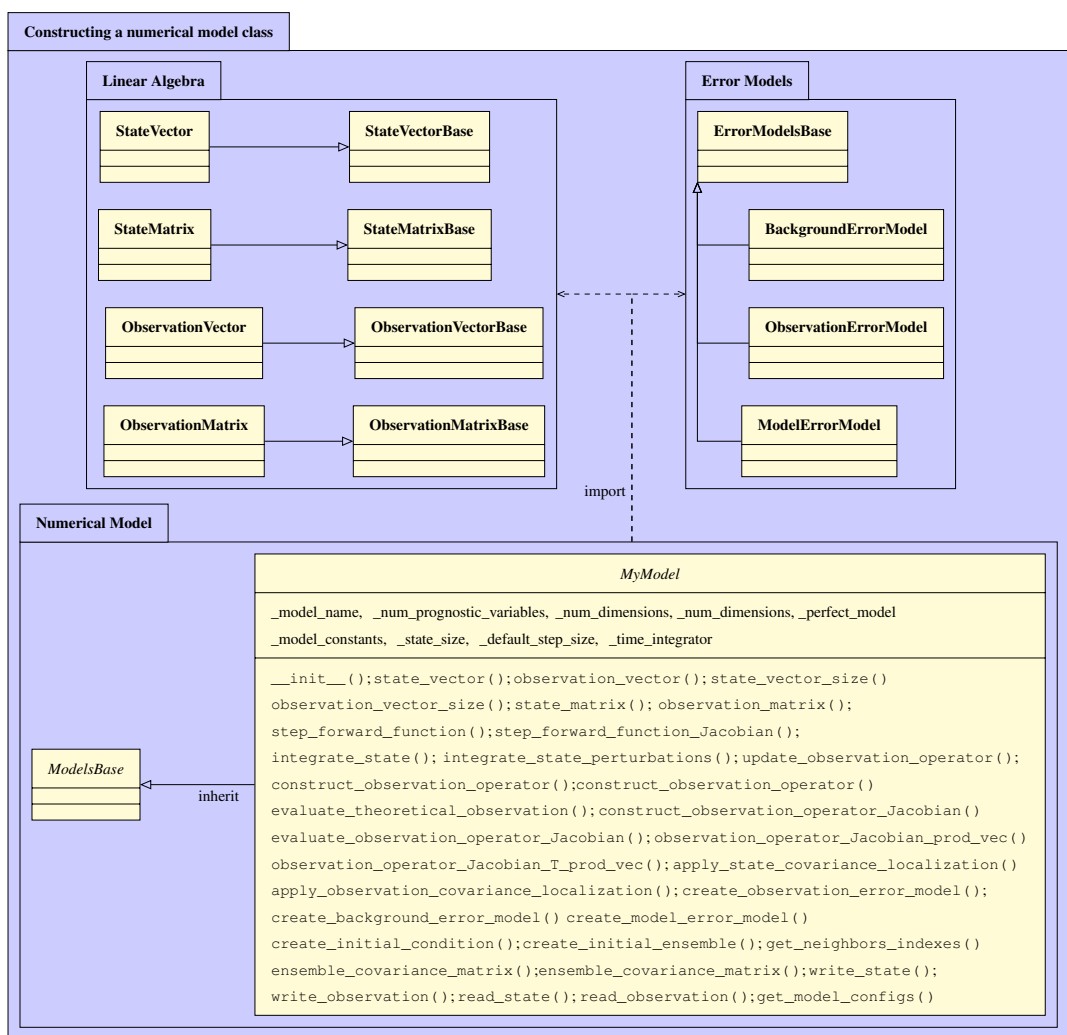

**Figure 17.** Illustration of a numerical model class named `MyModel`, and relations to the linear algebra and error models classes. A dashed arrow refers to an "*import*" relation, and a solid arrow represents an "*inherit*" relation.

To create a new class for DA filtering one derives it from the base class `FiltersBase`, imports appropriate routines, and defines the necessary functionalities. Note that each assimilation object has access to a model object, and consequently to the proper linear algebra data structures and associated functionalities through that model.

Unlike the base class for numerical models (`ModelsBase`), the filtering base class `FiltersBase` includes actual implementations of several widely used solvers. For example, an implementation of the method `FiltersBase.filtering_cycle()` is provided to carry out a single filtering cycle by applying a forecast phase followed by an analysis phase (or vice-versa, depending on stated configurations).

```
import dates_utility as utility
from models_base import ModelsBase
from state_vector_numpy import StateVectorNumpy as StateVector
from state_matrix_numpy import StateMatrixNumpy as StateMatrix
from state_matrix_sp_scipy import StateMatrixSpSciPy as SparseStateMatrix

class MyModel(ModelsBase):
    _model_name = "MyModel"
    _default_model_configs = dict(model_name=_model_name)

    def __init__(self, model_configs=None, output_configs=None):
        """ Constructor; MyModel class implementation. """
        # Aggregate passed configurations with default configurations
        model_configs = utility.aggregate_configurations(model_configs, DummyModel._default_model_configs)
        self.model_configs = utility.aggregate_configurations(model_configs, ModelsBase._default_model_configs)
        self._output_configs = utility.aggregate_configurations(output_configs, ModelsBase._default_output_configs)

    def state_vector(self):
        """ initialize an empty state vector """
        return StateVector(np.zeros(self.state_size()))

    def state_matrix(self, create_sparse=False):
        """ initialize an dense/sparse empty state matrix """
        state_size = self.state_size()
        if create_sparse:
            return SparseStateMatrix(sparse.lil_matrix((state_size, state_size)))
        else:
            return StateMatrix(np.zeros((state_size, state_size)))
```

**Figure 18.** The leading lines of an implementation of a class for the model `MyModel` derived from the models base class `ModelsBase`. Linear algebra objects are derived from Numpy-based (or Scipy-based) objects.

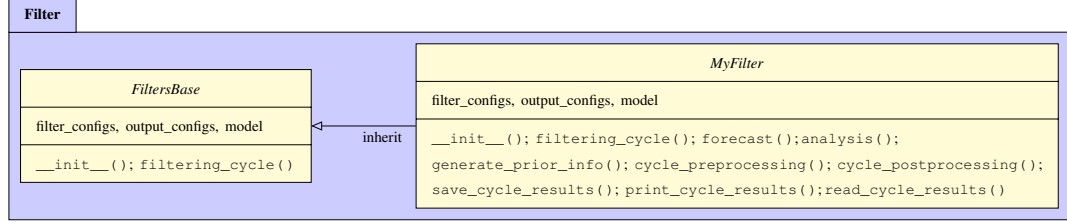

**Figure 19.** Illustration of a DA filtering class `MyFilter`, and its relation to the filtering base class. A solid arrow represents an "*inherit*" relation.

Figure 19 illustrates a filtering class named `MyFilter` that works by carrying out analysis and forecast steps in the ensemble-based statistical framework. Code Snippet in Figure 20 shows the leading lines of an implementation of the `MyFilter` class.

```python
import dates_utility as utility
from filters_base import FiltersBase
from models_base import ModelsBase

class MyFilter(FiltersBase):
    _filter_name = "MyFilter"
    _def_local_filter_configs = dict(model=None, filter_name=_filter_name)
    _local_def_output_configs = dict(scr_output=True, file_output=False,
                                     filter_statistics_dir='Filter_Statistics',
                                     model_states_dir='Model_States_Repository',
                                     observations_dir='Observations_Rpository')

    def __init__(self, filter_configs=None, output_configs=None):
        """ Constructor; MyFilter class implementation """
        err_msg = "A model object reference MUST be passed in 'filter_configs' as value to the key 'model'..."
        assert isinstance(filter_configs['model'], ModelsBase), err_msg

        # aggregate configurations, and attach filter_configs, output_configs to the filter object.
        filter_configs = utility.aggregate_configurations(filter_configs, MyFilter._def_local_filter_configs)
        output_configs = utility.aggregate_configurations(output_configs, MyFilter._local_def_output_configs)
        FiltersBase.__init__(filter_configs=filter_configs, output_configs=output_configs)
        self.model = self.filter_configs['model']

    def filtering_cycle(self):
        """ Carry out a single filtering cycle """
        FiltersBase.filtering_cycle()
        # Add further functionality if you wish...

    def forecast(self):
        """ Forecast step of the filter """
        #

    def analysis(self, *args, **kwargs):
        """ Analysis step of the filter """
        #
```

**Figure 20.** The leading lines of an implementation of a DA filter; the `MyFilter` class is derived from the filters base class `FiltersBase`.

## 6   Discussion and Concluding Remarks

This work describes DATeS, a flexible and highly-extensible package for solving data assimilation problems. DATeS seeks to provide a unified testing suite for data assimilation applications that allows researchers to easily compare different methodologies in different settings with minimal coding effort. The core of DATeS is written in Python. The main functionalities, such as

5   model propagation, filtering, and smoothing code, can however be written in high-performance languages such as C or Fortran to attain high levels of computational efficiency.

While, we introduced several assimilation schemes in this paper, the current version, DATeS v1.0, emphasizes the statistical assimilation methods. DATeS provide the essential infrastructure required to combine elements of a variational assimilation algorithm with other parts of the package. The variational aspects of DATeS, however, require additional work that includes

10   efficient evaluation of the adjoint model, checkpointing, and handling weak constraints. A new version of the package, under

development, will carefully address these issues, and will provide implementations of several variational schemes. The variational implementations will be derived from the 3D- and 4D-Var classes implemented in the current version (DATeS v1.0).

The current version of the package presented in this work, DATeS v1.0, can be situated between professional data assimilation packages such as DART, and simplistic research-grade implementations. DATeS is well-suited for educational purposes as a learning tool for students and new comers to the data assimilation research field. It can also help data assimilation researchers develop specific components of the data assimilation process, and easily use them with the existing elements of the package. For example, one can develop a new filter, and interface an existing physical model, and error models, without the need to understand how these components are implemented. This requires unifying the interfaces between the different components of the data assimilation process, which is an essential feature of DATeS. These features allow for optimal collaboration between teams working on different aspects of a data assimilation system.

To contribute to DATeS, by adding new implementations, one must comply to the naming conventions given in the base classes. This requires building proper Python interfaces for the implementations intended to be incorporated with the package. Interfacing operational models, such the weather research and forecasting model (WRF) (Skamarock et al., 2005), in the current version, DATeS v1.0, is expected to require substantial work. Moreover, DATeS does not yet support parallelization, which limits its applicability in operational settings.

The authors plan to continue developing DATeS with the long-term goal of making it a complete data assimilation testing suite that includes support for variational methods, as well as interfaces with complex models such as quasi-geostrophic global circulation models. Parallelization of DATeS, and interfacing large-scale models such as the weather research and forecasting model (WRF) (Skamarock et al., 2005), will also be considered in the future.

*Code and data availability.* The code of DATeS-v1.0 is available from https://doi.org/10.5281/zenodo.1247464. The online documentation, and alternative download links are available at http://people.cs.vt.edu/~attia/DATeS/index.html.

*Competing interests.* The authors declare that they have no conflict of interest.

*Acknowledgements.* The authors would like to thank Mahesh Narayanamurthi, Paul Tranquilli, Ross Glandon, and Arash Sarshar from the Computational Science Laboratory (CSL) at Virginia Tech, and Vishwas Rao from Argonne National Laboratory, for their contributions to an initial version of DATeS. This work has been supported in part by awards NSF CCF-1613905, NSF ACI–1709727, AFOSR DDDAS 15RT1037, and by the Computational Science Laboratory at Virginia Tech.

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
