# Peer review of "DATeS: A Highly-Extensible Data Assimilation Testing Suite v1.0"

_Geoscientific Model Development, 2018_

## Short Comment (SC1) · 4 Apr 2018

Dear authors,

in my role as Executive editor of GMD, I would like to bring to your attention our Editorial version 1.1: http://www.geosci-model-dev.net/8/3487/2015/gmd-8-3487-2015.html This highlights some requirements of papers published in GMD, which is also available on the GMD website in the 'Manuscript Types' section: http://www.geoscientific-model-development.net/submission/manuscript_types.html In particular, please note that for your paper, the following requirement has not been met in the Discussions paper:

- "The main paper must give the model name and version number (or other unique

identifier) in the title."

Please provide the version number of DATeS in the title of your revised manuscript.

As explained in https://www.geoscientific-model-development.net/about/manuscript_types.html GMD is encouraging authors to upload the program code of models (including relevant data sets) as supplement or make the code and data of the exact model version described in the paper accessible through a DOI (digital object identifier). In case your institution does not provide the possibility to make electronic data accessible through a DOI you may consider other providers (eg. zenodo.org of CERN) to create a DOI. Please note that in the code accessibility section you can still point the reader to how to obtain the newest version. If for some reason the code and/or data cannot be made available in this form (e.g. only via e-mail contact) the "Code Availability" section need to clearly state the reasons for why access is restricted (e.g. licensing reasons).

Yours, Astrid Kerkweg

---

## Referee Comment (RC1) · Anonymous Referee #1 · 10 May 2018

Review of GMD-2018-30-1

DATeS is aiming to provide a flexible and highly-extensible data assimilation testing suite.

It seeks to provide a unified testing suite for data assimilation applications allowing data assimilation researchers to easily compare different methodologies in different settings with minimal coding effort.

The core of DATeS is written in Python. The main functionalities, such as model propagation, filtering, and smoothing code, can be written in low-level languages such as C or Fortran to attain high levels of computational efficiency.

It allows for easy interfacing with external third-party code written in various languages,

e.g., linear algebra routines written in Fortran, analysis routines written in Matlab, or "forecast" models written in C. This should help the researchers to focus their energy on implementing and testing their own analysis algorithms.

The DATeS architecture abstracts the following generic components of any DA system:

1. linear algebra routines,

2. a "forecast" computer model that includes the discretization of the physical processes,

3. error models, e.g. observation and background error,

4. an analysis methodology, e.g., a filter or a smoother.

At this stage the authors tested it on several popular test models for data assimilation:

(i) lorenz_models.Lorenz3: A class implementing the 3-variables Lorenz model Lorenz (1963).

(ii) lorenz_models.Lorenz96: An implementation of the Lorenz96 model Lorenz (1996).

(iii) cartesian_swe_model.CartesianSWE: Cartesian shallow-water equations model Gustafsson (1971); Navon and De-Villiers (1986) written in C, with a SWIG wrapper.

15 (iv) qg_1p5_model.QG1p5: Quasi-geostrophic (QG)model with double-gyre wind forcing and bi-harmonic friction Sakov and Oke (2008) written in Fortran, with a F2Py wrapper.

DATeS provides the following classes for several versions of the EnKF, the HMC family of filters, and a vanilla implementation of the particle filter:

The authors plan to continue developing DATeS with the long-term goal of making it a complete data assimilation testing suite that includes support for variational methods, as well as interfaces with complex.

My main issues to address regarding this interesting contribution are:

1. The paper has the character of a user manual and a missing essential part is that of highlighting in detail the difficulties, both technical and theoretical to use a full physics 4- D Var operational model and its adjoint in DATeS framework.

2. They should mention and discuss use of minimization algorithms to minimize the cost functional in particular for non-differentiable optimization.See Steward et al (2012,2014)

3 How to implement state of the art data assimilation methods for high-dimensional non-Gaussian problems.

See Vetra-Carvalho et al (2018)

4. How to deal with check-pointing and incremental 4-D VAR in the framework of DaTeS.

5. How to deal with model error in weal constraint data assimilation

6.The authors should present advantages and shortcomings of DATeS as well as expectations of benefits of its use.

Due to its perceived useful application I recommend the paper be accepted subject to addressing these medium/minor revisions.

References

J. L. Steward, I.M.Navon,M. Zupanski and N. Karmitsa:Impact of Non-Smooth Observation Operators on Variational and Sequential Data Assimilation for a Limited-Area Shallow Water Equations Model.Quart. Jour. Roy. Met Soc. , Volume 138, Issue 663, 323–339, January 2012, Part B (2012)

Jeff L Steward, Ionel Michael Navon, Napsu Karmitsa, and Milija Zupanski : User Manual for code of the paper Impact of Non-Smooth Observation Operators on Variational and Sequential Data Assimilation for a Limited-Area Shallow Water Equations Model in QJRMS (2014)

Sanita Vetra-Carvalho, Peter Jan van Leeuwen, Lars Nerger, Alexander Barth, M. Umer Altaf, Pierre Brasseur, Paul Kirchgessner & Jean-Marie Beckers (2018):State-of-the art stochastic data assimilation methods for high-dimensional non-Gaussian problems, Tellus A: Dynamic Meteorology and Oceanography, 70:1, 1445364, DOI: 10.1080/16000870.2018.1445364
* * *

---

## Referee Comment (RC2) · K. Law (Referee) · 18 May 2018

The authors and the first referee have done a very good job of summarising the contribution of this paper. I second the very relevant requests of the first referee, and I would like to additionally raise a few other points.

Perhaps the most important point to touch on here is the potential applicability of DATeS as a benchmarking tool. The most challenging aspect of developing community-wide benchmarks might be the abundance of tuning parameters required to get "optimal" performance. Please discuss this point, and initialise some default cases. For example, if you have Lorenz 63 with canonical parameters, observations every h time steps, and a given observational noise and prior, then the "best" you can do with EnKF is

using X set of tuning parameters. The best you can do with 4DVAR is with Y set of tuning parameters. And so on... Of course one has to define an error metric, and optimality of tuning parameters will presumably change, but something like RMSE and rank histogram of the innovation (or truth in the case of simulated data) is a good start. As the software is used this can then be updated by the community. When a new algorithm is proposed it can then be sized up against the community-endorsed benchmarks. This is of course asking a lot from both the software and the community, but it is good to aim high. Such benchmarking tool is needed and would be very useful.

Also, if you are aiming for the posterior distribution in a general non-Gaussian case, for example as a benchmark against which to evaluate other algorithms, or to compute higher moments, multiple modes, or tail probabilities, then the model and tuning parameters can again be recorded and the results can be challenged (for example, in case a mode has been missed) as new and improved methodology is introduced for general posterior inference and as computers getter bigger, stronger, and faster. This approach can in principle manifest reproducible, evolving, and community-endorsed gold-standard benchmarks, which can be used in addition to such metrics as RMSE and rank histograms in vetting existing and new algorithms in various scenarios.

Specific comments:

* p2, line 15: as a gateway for someone new to the field or interested in learning about the methodology without all the complexities, one could also mention some extremely simple software, for example the pedagogical applied mathematics reference on data assimilation Law etal 2015 provides a concise set of codes including examples of many of the modern data assimilation algorithms distilled to the level that they are single Matlab scripts of fewer than 50 lines which run in a fraction of a second.

http://tiny.cc/damat

* Sec 2: this presentation is not quite complete. It needs to be considerably cleaned up and made complete. For example, you do not state how the model enters the

picture: is it the case that there is a single state $x_k^a$, for analysis, and $x_{k+1}^b = \mathcal{M}_{k,k+1}(x_k^a)$ ? You do not define $\hat{}^{\rm true}$, and it vanishes in (3). Do you need it?

* p4, line 8: minimum variance estimator: even if the Gaussian background assumption holds, and you have an infinite ensemble, this is not the minimum variance estimator, which is the posterior mean. It is the minimum variance estimator among those which are linear in the observation. This does not include the posterior mean in case $\mathcal{H}_k$ is nonlinear. Same comment holds a the end of p4, relating to smoothing.

* p5, line 17: "not generally efficient" -> "not generally considered to be efficient". A lot of recent work has illustrated the potential applicability of particle filters for high-dimensional problems, although they have not yet been used operationally. In addition to the reference suggested by the other referee, see below references by Crisan, Jasra, van Handel, Poterjoy, Potthast, and van Leeuwen.

* p5, around line 20: it is important to point out here that MCMC is generally applicable only to static problems, i.e. a single posterior distribution for a single window of observations, with a known prior/background. Once you step forward to the next window, you will not in general have a closed form or even a good approximation for the new prior/background (and it will most likely not be Gaussian). One needs to therefore be careful when applying MCMC in a recursive/filtering context. I'm not familiar with the cited papers although I'm sure they deal with this in a sensible way. But it needs to be mentioned here. And, nonetheless MCMC methods are useful as a benchmarking tool for a single assimilation window with a known background – see Law and Stuart 2012.

* p7, line 6: "It is common in DA applications to assume a perfect forecast model, a case where the model is deterministic rather than stochastic." Perhaps you mean just "It is common in DA applications to assume the model is deterministic rather than stochastic." ? Here inflation will of course be needed for ensemble methods in order

Interactive
comment

to have stability, and in any case the background covariance incorporates some sort of model error, so I don't see this as a perfect model scenario.

\* Related to the above point, localisation and inflation should be mentioned explicitly, due to their key roles in the field. Presumably they need to be incorporated within the analysis algorithm component.

\*p8, line 14: state-size square matrices! Surely this is limited to very small problems, so some discussion of low-rank approximation of these matrices is important, and to be robust the code should not even try to construct these as full matrices if the state is high-dimensional.

\* p14: define DEnKF

Beskos, Alexandros, et al. "A stable particle filter for a class of high-dimensional state-space models." Advances in Applied Probability 49.1 (2017): 24-48.

Law, Kody JH, and Andrew M. Stuart. "Evaluating data assimilation algorithms." Monthly Weather Review 140.11 (2012): 3757-3782.

Law, Kody, Andrew Stuart, and Konstantinos Zygalakis. Data assimilation: a mathematical introduction. Vol. 62. Springer, 2015.

Llopis, Francesc Pons, et al. "Particle Filtering for Stochastic Navier-Stokes Signal Observed with Linear Additive Noise." arXiv preprint arXiv:1710.04586 (2017).

Poterjoy, Jonathan. "A localized particle filter for high-dimensional nonlinear systems." Monthly Weather Review 144.1 (2016): 59-76.

Potthast, Roland, et al. "A Localised Adaptive Particle Filter within an Operational NWP Framework." (2017).

Rebeschini, Patrick, and Ramon Van Handel. "Can local particle filters beat the curse of dimensionality?." The Annals of Applied Probability 25.5 (2015): 2809-2866.

Ades, Melanie, and Peter J. Van Leeuwen. "The equivalent‐weights particle filter in a high‐dimensional system." Quarterly Journal of the Royal Meteorological Society 141.687 (2015): 484-503.

---

## Referee Comment (RC3) · Anonymous Referee #3 · 22 May 2018

General comments

I fully agree with the comments of the other two referees. In particular, I would like to echo Referee #1 specific comment on the need to improve scientific/technical aspect of the paper, so that it reads more like a scientific paper rather than a user manual. I also find Referee #2 comments on covariance localization and limitations due to using state-size square matrices very important to address.

Specific comments

1. What is the largest state size that DATeS can handle, given that a full-state quadratic matrix is used ($Nstate \times Nstate$), as stated on page 8, line 15 and in Figure 13? On page 14, line 7, you mentioned that the state size was $Nstate=16641$ in your QG model

experiment. Did you really evaluate matrices of the size 16641 x 16641 in that experiment? Please elaborate on how high-dimensional model state problems are addressed and what the limitations of the current algorithm are regarding this aspect. Furthermore, an option to use non-quadratic state matrices of size Nstate x Nens needs to be discussed as a possible extension in the future, explaining how this can be done.

2. Are applications of DATeS intended for complex atmospheric models, such as WRF or NOAA future operational global model (FV3, https://www.gfdl.noaa.gov/fv3/)? What about coupled atmosphere-ocean models, requiring coupled forecast error covariance and covariance localization? Please elaborate on these issues.

3. How extensible the DATeS is with respect to the following aspects: (1) Weak constraint ensemble, variational and hybrid ensemble-variational data assimilation (also pointed out by Referee #1) and (2) Data assimilation problems with correlated observation errors (e.g., addressing cross-channel correlations of satellite radiances).

4. A special attention needs to be taken to forecast error covariance localization for ensemble-based approaches (also pointed out by Referee #2). Two covariance localization methods are typically used: (1) localization in model space and (2) localization in observation space. Please explain, in more detail, the covariance localization approach used in the current version of DATeS and how you plan to extend it in future applications with more complex models.

5. Page 3, Eq. (3): The conditions should also include an assumption that the observation and background errors are independent (un-correlated).

---

## Referee Comment (RC4) · Anonymous Referee #4 · 23 May 2018

Thanks to the first reviewer who has provided a great summary of the paper contributions. I also agree that the current draft is more like a user manual than a technical paper. A simplified language use and less jargon may improve the readability in the introduction part.

1. The author could mention the educational purpose of DATeS in the very beginning.

2. Pg2 Line 5, please add reference for DART applications.

3. Starting Pg4 Line 5, the authors introduced DA in general, then at Pg 5, line 22, the authors mentioned they have implemented several flavors of the above mentioned DA schemes. It would be nice to mention which versions are implemented and which are not. Alternatively, the authors could start with the summary of DATeS's implementation,

then provide details of each implementation.

4. The authors have mentioned in several places that Fortran and C are low-level languages. They are not. Low-level languages are generally referred to machine language and assembly language.

5. The authors are encouraged to develop a Graphic User Interface (GUI) for flexibility and ease of use.

6. The DA software and programing language comparison on pg 2 can be formatted better in terns of the software cost, learning curve, ease of access, extensibility etc. to bring out the "unified" nature of the Python implementation.

7. The authors mentioned to compare different DA schemes, it would be nice to include an example of DA results comparison with DATeS.

8. I also agree with another referee that limitations of DATeS should be pointed out.
* * *

---

## Author Comment (AC1) · 4 Jul 2018

**DATeS : A Highly-Extensible Data Assimilation Testing Suite v1.0**

Ahmed Attia and Adrian Sandu

Dear Editor,

Please find attached the reviewed version of the manuscript "DATeS : A Highly-Extensible Data Assimilation Testing Suite v1.0" by A. Attia and A. Sandu.

We thank the reviewers for their useful feedback. A point-wise list of responses to each of the reviewers' comments is attached.

Regards,

Ahmed Attia

**Technical Editor: Astrid Kerkweg.** Dear authors,

In my role as Executive editor of GMD, I would like to bring to your attention our Editorial version 1.1: http://www.geosci-model-dev.net/8/3487/2015/gmd-8-3487-2015.html This highlights some requirements of papers published in GMD, which is also available on the GMD website in the 'Manuscript Types' section: http://www.geoscientific-modeldevelopment.net/submission/manuscript_types.html In particular, please note that for your paper, the following requirement has not been met in the Discussions paper:

- "The main paper must give the model name and version number (or other unique identifier) in the title."

Please provide the version number of DATeS in the title of your revised manuscript. As explained in https://www.geoscientific-model-development.net/about/manuscript_types.html GMD is encouraging authors to upload the program code of models (including relevant data sets) as supplement or make the code and data of the exact model version described in the paper accessible through a DOI (digital object identifier). In case your institution does not provide the possibility to make electronic data accessible through a DOI you may consider other providers (eg. zenodo.org of CERN) to create a DOI. Please note that in the code accessibility section you can still point the reader to how to obtain the newest version. If for some reason the code and/or data cannot be made available in this form (e.g. only via e-mail contact) the "Code Availability" section need to clearly state the reasons for why access is restricted (e.g. licensing reasons).

Answer: Thank you.

- We added v1.0 to the title of the paper.

- We created a DOI for the package, and updated the citations accordingly.
**Referee 1.** DATeS is aiming to provide a flexible and highly-extensible data assimilation testing suite. It seeks to provide a unified testing suite for data assimilation applications allowing data assimilation researchers to easily compare different methodologies in different settings with minimal coding effort. The core of DATeS is written in Python. The main functionalities, such as model propagation, filtering, and smoothing code, can be written in low-level languages such as C or FORTRAN to attain high levels of computational efficiency. It allows for easy interfacing with external third-party code written in various languages, e.g., linear algebra routines written in FORTRAN, analysis routines written in MATLAB, or "forecast" models written in C. This should help the researchers to focus their energy on implementing and testing their own analysis algorithms. The DATeS architecture abstracts the following generic components of any DA system:

1. linear algebra routines,

2. a "forecast" computer model that includes the discretization of the physical processes,

3. error models, e.g. observation and background error,

4. an analysis methodology, e.g., a filter or a smoother.

At this stage the authors tested it on several popular test models for data assimilation:

(i) lorenz_models.Lorenz3: A class implementing the 3-variables Lorenz model Lorenz (1963).

(ii) lorenz_models.Lorenz96: An implementation of the Lorenz96 model Lorenz (1996).

(iii) cartesian_swe_model.CartesianSWE: Cartesian shallow-water equations model Gustafsson (1971); Navon and De-Villiers (1986) written in C, with a SWIG wrapper.

(iv) qg_1p5_model.QG1p5: Quasi-geostrophic (QG)model with double-gyre wind forcing and bi-harmonic friction Sakov and Oke (2008) written in FORTRAN, with a F2Py wrapper.

DATeS provides the following classes for several versions of the EnKF, the HMC family of filters, and a vanilla implementation of the particle filter: The authors plan to continue developing DATeS with the long-term goal of making it a complete data assimilation testing suite that includes support for variational methods, as well as interfaces with complex.

Answer: Thank you, for the summary, and the insightful comments.

**My main issues to address regarding this interesting contribution are:**

1. The paper has the character of a user manual and a missing essential part is that of highlighting in detail the difficulties, both technical and theoretical to use a full physics 4D-Var operational model and its adjoint in DATeS framework.

   Answer:

   - We agree that the first version of the manuscript has the character of a user manual. Our intention was to balance between a research paper, and a technical description of the DATeS framework. We rewrote some parts of the paper to make it more like a research article, rather than a user manual. We hope the revised manuscript addresses this issue.

   - We completely agree on the importance of the variational approach to DA, as well as hybrid methods. Please see the answers to the following questions; we hope these will justify the current state of the variational aspects of DATeS .

2. They should mention and discuss use of minimization algorithms to minimize the cost functional in particular for non-differentiable optimization. See Steward et

al (2012,2014) Answer: We agree that an optimizer is an essential element of a variational data assimilation scheme. However, as highlighted in the article and especially in the reviewed manuscript, the design of DATeS makes each of its elements independent. This version of the package, version 1.0, contains the gluing components that enables adding variational schemes, following the same procedure as adding assimilation filters. The implementation of the variational scheme, however, requires much more effort due to the need to developing adjoint model, and handling forward and backward propagation especially in the presence of model errors. The optimization algorithm used to minimize the objective function, e.g. the negative posterior-log, can be an of-the-shelf product, or tailored for the algorithm in hand. As a proof of concept, we designed a 3D-Var and 4D-Var assimilation classes. Scipy out-of-the-box optimizers are used to minimize the objective function. The tangent linear model is designed in the model class, and the adjoint is automatically evaluated using FATODE following a checkpointing approach. Many variants of the 4D-Var assimilation scheme exist, including incremental 4D-Var, however, we opted to focus on filtering algorithms in this version of the package. The next version in development, DATeS V2.0, will highlight the variational aspects of the package, and will give access to several variational classes.

3. How to implement state of the art data assimilation methods for high-dimensional non-Gaussian problems. See Vetra-Carvalho et al (2018) Answer: As highlighted in the revised manuscript, the initial version of DATeS (v1.0) is not meant to provide implementations of all state of the art DA algorithms. In fact, with continuous advances of DA methodologies, we don't believe this is a task that should be carried out by a single team of developers. Our main goal is to make DATeS available as extensible testing environment, where researchers can add efficient and accurately-tuned implementations of their own algorithms, DATeS however, provides an initial seed with example implementations, those could be discussed,

and enhanced by the ever-growing community of DA researchers and experts.

4. How to deal with check-pointing and incremental 4-D VAR in the framework of DATeS . Answer: As a proof of concept, we added an implementation of the 4D-Var problem which explains checkpointing, however we defer the detailed discussion of the variational aspects of DATeS to the upcoming version of the package, which is currently under development.

5. How to deal with model error in weak constraint data assimilation Answer: DATeS is meant to be a gluing package to existing and new implementation, where each of the abstracted components can be implemented independently. This includes forecast models, error models, and assimilation procedures. Forecast model errors are abstracted and can be used by different forecast models in DATeS . The approach followed for handling model errors in the weak constraint formulation depends on the assimilation algorithm, and is completely independent from the other parts of the package. Again, we would like to stress that in the next version of DATeS currently under development, we will provide several variants of the 4D-Var scheme, and we intend to give attention to the weak constraint formulation. This is now highlighted in Section 6.

6. The authors should present advantages and shortcomings of DATeS as well as expectations of benefits of its use. Answer: A discussion has been added to the conclusion, Section 6.

**Due to its perceived useful application I recommend the paper be accepted subject to addressing these medium/minor revisions.**

**References**

1. J. L. Steward, I.M.Navon,M. Zupanski and N. Karmitsa:Impact of Non-Smooth Observation Operators on Variational and Sequential Data Assimilation for a

Limited-Area Shallow Water Equations Model.Quart. Jour. Roy. Met Soc. , Volume 138, Issue 663, 323–339, January 2012, Part B (2012)

2. Jeff L Steward, Ionel Michael Navon, Napsu Karmitsa, and Milija Zupanski : User Manual for code of the paper Impact of Non-Smooth Observation Operators on Variational and Sequential Data Assimilation for a Limited-Area Shallow Water Equations Model in QJRMS (2014)

3. Sanita Vetra-Carvalho, Peter Jan van Leeuwen, Lars Nerger, Alexander Barth, M. Umer Altaf, Pierre Brasseur, Paul Kirchgessner & Jean-Marie Beckers (2018):State-of-the art stochastic data assimilation methods for high-dimensional non-Gaussian problems, Tellus A: Dynamic Meteorology and Oceanography, 70:1, 1445364, DOI: 10.1080/16000870.2018.1445364

[Figure]

**Referee 2: K. Law** The authors and the first referee have done a very good job of summarising the contribution of this paper. I second the very relevant requests of the first referee, and I would like to additionally raise a few other points. Perhaps the most important point to touch on here is the potential applicability of DATeS as a benchmarking tool. The most challenging aspect of developing community-wide benchmarks might be the abundance of tuning parameters required to get "optimal" performance. Please discuss this point, and initialise some default cases. For example, if you have Lorenz 63 with canonical parameters, observations every h time steps, and a given observational noise and prior, then the "best" you can do with EnKF is using X set of tuning parameters. The best you can do with 4DVAR is with Y set of tuning parameters. And so on... Of course one has to define an error metric, and optimality of tuning parameters will presumably change, but something like RMSE and rank histogram of the innovation (or truth in the case of simulated data) is a good start. As the software is used this can then be updated by the community. When a new algorithm is proposed it can then be sized up against the community-endorsed benchmarks. This is of course asking a lot from both the software and the community, but it is good to aim high. Such benchmarking tool is needed and would be very useful. Also, if you are aiming for the posterior distribution in a general non-Gaussian case, for example as a benchmark against which to evaluate other algorithms, or to compute higher moments, multiple modes, or tail probabilities, then the model and tuning parameters can again be recorded and the results can be challenged (for example, in case a mode has been missed) as new and improved methodology is introduced for general posterior inference and as computers get bigger, stronger, and faster. This approach can in principle manifest reproducible, evolving, and community-endorsed gold-standard benchmarks, which can be used in addition to such metrics as RMSE and rank histograms in vetting existing and new algorithms in various scenarios.

Answer: We added Section 4.4 that discusses the basic benchmarking tools, including RMSE, rank histograms, and rank histogram uniformity measures. We find it more helpful for researchers to be able to easily, and quickly produce benchmarking results, given the code, rather than publishing results based on our runs. Following the referee suggestion, we added to DATeS v1.0, a few scripts explaining how to create benchmark experiments to be challenged by new implementations, for example using Lorenz96 model. It is worth mentioning that the goal of DATeS is not to promote a specific setup, such as unimodal posteriors. DATeS provides the main infrastructure that makes is possible to add various components, without worrying about compatibility with the other parts of the package. DATeS provides several basic, and advanced MCMC classes suitable for Gaussian, and non-Gaussian settings. Visual and analytic tools for assessing coverage of the sample to a multimodal distribution is indeed of utmost importance. These tools however, are utility tools, that can be easily added to the package by the developers or the community.

**Specific comments:**

- p2, line 15: as a gateway for someone new to the field or interested in learning about the methodology without all the complexities, one could also mention some extremely simple software, for example the pedagogical applied mathematics reference on data assimilation Law etal 2015 provides a concise set of codes including examples of many of the modern data assimilation algorithms distilled to the level that they are single Matlab scripts of fewer than 50 lines which run in a fraction of a second. http://tiny.cc/damat

  Answer: Thank you, for pointing us out to the text. We mentioned it in the text, and added citation accordingly.

- Sec 2: this presentation is not quite complete. It needs to be considerably cleaned up and made complete. For example, you do not state how the model enters the picture: is it the case that there is a single state $x_k a$, for analysis, and $x_{k+1} b = \mathcal{M}_{k,k+1}(x_k a)$ ? You do not define true, and it vanishes in (3). Do you need it?

  Answer: We define the model error in terms of the true model state $\mathbf{x}true$, to give

the readers an idea of error distributions. The model $\mathcal{M}$ is mainly used in the forecast step of an assimilation algorithm. This was initially mentioned, however, we updated the text in this section to make it more clear. We hope the updated text reads better, and cover the basics of DA algorithms.

- p4, line 8: minimum variance estimator: even if the Gaussian background assumption holds, and you have an infinite ensemble, this is not the minimum variance estimator, which is the posterior mean. It is the minimum variance estimator among those which are linear in the observation. This does not include the posterior mean in case $\mathcal{H}_k$ is nonlinear. Same comment holds a the end of p4, relating to smoothing.

  Answer: We agree. Text updated.

- p5, line 17: "not generally efficient" -> "not generally considered to be efficient". A lot of recent work has illustrated the potential applicability of particle filters for high dimensional problems, although they have not yet been used operationally. In addition to the reference suggested by the other referee, see below references by Crisan, Jasra, van Handel, Poterjoy, Potthast, and van Leeuwen.

  Answer: Text updated, and citation added.

- p5, around line 20: it is important to point out here that MCMC is generally applicable only to static problems, i.e. a single posterior distribution for a single window of observations, with a known prior/background. Once you step forward to the next window, you will not in general have a closed form or even a good approximation for the new prior/background (and it will most likely not be Gaussian). One needs to therefore be careful when applying MCMC in a recursive/filtering context. I'm not familiar with the cited papers although I'm sure they deal with this in a sensible way. But it needs to be mentioned here. And, nonetheless MCMC methods are useful as a benchmarking tool for a single assimilation window with a known background – see Law and Stuart 2012.

Answer: We agree that applying MCMC requires accurate representation of the prior, which is generally intractable in sequential nonlinear settings. This issue of non-Gaussianity is mentioned in the updated text, and is highlighted both in the introduction and the new section on Benchmarking.

- p7, line 6: "It is common in DA applications to assume a perfect forecast model, a case where the model is deterministic rather than stochastic." Perhaps you mean just "It is common in DA applications to assume the model is deterministic rather than stochastic." ? Here inflation will of course be needed for ensemble methods in order to have stability, and in any case the background covariance incorporates some sort of model error, so I don't see this as a perfect model scenario.

  Answer: We agree. Text updated.

- Related to the above point, localisation and inflation should be mentioned explicitly, due to their key roles in the field. Presumably they need to be incorporated within the analysis algorithm component.

  Answer: Indeed inflation and localization are critically important for ensemble-based assimilation algorithms. We prefer to abstract tools and functions, common to assimilation methods, such as inflation and covariance localization.. For that, we consider inflation and localization to be utility procedures that can be called by any new ensemble-based filter. The utility module in DATeS provide methods to carry out these procedures in different modes, including state space and observation space localization. We also support both space-time dependent covariance localization. These facts were highlighted in the revised version of the manuscript.

- p8, line 14: state-size square matrices! Surely this is limited to very small problems, so some discussion of low-rank approximation of these matrices is important, and to be robust the code should not even try to construct these as full matrices if the state is high-dimensional.

Answer: We agree. A note is added to the text.

- p14: define DEnKF

Answer: DEnKF stands for "Deterministic Ensemble Kalman Filter". This is clarified in the text.

**References**

1. Beskos, Alexandros, et al. "A stable particle filter for a class of high-dimensional statespace models." Advances in Applied Probability 49.1 (2017): 24-48.

2. Law, Kody JH, and Andrew M. Stuart. "Evaluating data assimilation algorithms." Monthly Weather Review 140.11 (2012): 3757-3782.

3. Law, Kody, Andrew Stuart, and Konstantinos Zygalakis. Data assimilation: a mathematical introduction. Vol. 62. Springer, 2015.

4. Llopis, Francesc Pons, et al. "Particle Filtering for Stochastic Navier-Stokes Signal Observed with Linear Additive Noise." arXiv preprint arXiv:1710.04586 (2017). Poterjoy, Jonathan. "A localized particle filter for high-dimensional nonlinear systems." Monthly Weather Review 144.1 (2016): 59-76.

5. Potthast, Roland, et al. "A Localised Adaptive Particle Filter within an Operational NWP Framework." (2017). Rebeschini, Patrick, and Ramon Van Handel. "Can local particle filters beat the curse of dimensionality?." The Annals of Applied Probability 25.5 (2015): 2809-2866.

6. Ades, Melanie, and Peter J. Van Leeuwen. "The equivalent-weights particle filter in a high-dimensional system." Quarterly Journal of the Royal Meteorological Society 141.687 (2015): 484-503.

**Referee 3. General Comments:**

I fully agree with the comments of the other two referees. In particular, I would like to echo Referee #1 specific comment on the need to improve scientific/technical aspect of the paper, so that it reads more like a scientific paper rather than a user manual. I also find Referee #2 comments on covariance localization and limitations due to using state-size square matrices very important to address.

Answer: Thank you. We hope the updated version, and the responses to the comments made by Referee #1 and Referee #2 properly address these concerns.

**Specific Comments:**

1. What is the largest state size that DATeS can handle, given that a full-state quadratic matrix is used $N_{state} \times N_{state}$, as stated on page 8, line 15 and in Figure 13? On page 14, line 7, you mentioned that the state size was $N_{state} = 16641$ in your QG model experiment. Did you really evaluate matrices of the size $16641 \times 16641$ in that experiment? Please elaborate on how high-dimensional model state problems are addressed and what the limitations of the current algorithm are regarding this aspect. Furthermore, an option to use non-quadratic state matrices of size $N_{state} \times N_{ens}$ needs to be discussed as a possible extension in the future, explaining how this can be done.

    Answer: DATeS provide both dense and sparse implementations of state- and observation-size matrices for experimental purposes. Low-rank approximations, sparse data structures, and matrix-free algorithms should be considered for large-scale settings. We highlighted these facts in the reviewed manuscript. We also, discussed the need to constructing non-quadratic state matrices of size $N_{state} \times N_{ens}$, and discussed the approach DATeS follows in handling such matrices. Specifically, we represent state ensembles as list of states, and provide utility functions that, given an ensemble of states, can calculate ensemble statistics, including variances, and covariance trace. Moreover, the linear algebra operations

such as matrix-vector product that involves a matrix representation of ensembles of states are implemented as implemented following a matrix-free approach for efficiency. These facts have been highlighted in the revised manuscript.

2. Are applications of DATeS intended for complex atmospheric models, such as WRF or NOAA future operational global model (FV3, https://www.gfdl.noaa.gov/fv3/)? What about coupled atmosphere-ocean models, requiring coupled forecast error covariance and covariance localization? Please elaborate on these issues.

   Answer:

   • As mentioned in the introduction, Section 1, DATeS is intended to be positioned between the simple typical research-grade implementations and the professional implementation of DART, but with the capability to utilize large physical models. The current version of DATeS (v1.0) does not support parallelization, and would require a lot of work, and effort to interface operational models such as WRF. One effort being currently pursued is to rebuild DATeS on top of Argonne's PETSc, which would be more suited to interface operational models such as WRF, and FV3.

   • Covariance localization and inflation are discussed in more details in the revised manuscript.

3. How extensible the DATeS is with respect to the following aspects: (1) Weak constraint ensemble, variational and hybrid ensemble-variational data assimilation (also pointed out by Referee #1) and (2) Data assimilation problems with correlated observation errors (e.g., addressing cross-channel correlations of satellite radiances).

   Answer:

   • As indicated in the response to Referee #1, the approach followed to handle weak constraint formulation controls the implementation of the assimilation

algorithm, and does not pose limitations on the other parts of the package. It may require access to a model error model, which is accessible through the forecast error object, and is independent from the specific implementation of the assimilation class itself. The current version DATeS v1.0, provides implementation of strong constraint 4D-Var as a proof of concept. However, we defer the discussion of the variational aspects of the package, which is under development, to future releases.

- Observation error correlations are part of the observation error model. The Gaussian error models provided by DATeS support both correlated and un-correlated errors, and construct the covariance matrices accordingly. The covariance matrices are stored in appropriate sparse formats, unless a dense matrix is explicitly requested. Since these covariance matrices are either state or observation matrices, they provide access to all proper linear algebra routines. This means that, the code written with access to an observation error model, and its components should work for both correlated and uncorrelated observations. This is now highlighted in the revised manuscript.

4. A special attention needs to be taken to forecast error covariance localization for ensemble-based approaches (also pointed out by Referee #2). Two covariance localization methods are typically used: (1) localization in model space and (2) localization in observation space. Please explain, in more detail, the covariance localization approach used in the current version of DATeS and how you plan to extend it in future applications with more complex models.

Answer: Both space, and observation localization are available in DATeS . We also support space-time dependent inflation and localization. Covariance localization and inflation are discussed in more details in the revised manuscript.

5. Page 3, Eq. (3): The conditions should also include an assumption that the

observation and background errors are independent (un-correlated).

Answer: A note is added to the text with the assumption that observation error and background errors are uncorrelated.

[Figure]

**Referee 4.** Thanks to the first reviewer who has provided a great summary of the paper contributions. I also agree that the current draft is more like a user manual than a technical paper. A simplified language use and less jargon may improve the readability in the introduction part.

Answer: Thank you. We hope the updated version, and the responses to the comments made by Referee #1 and Referee #3 properly address these concerns.

1. The author could mention the educational purpose of DATeS in the very beginning.

   Answer: A remark explaining educational purpose of DATeS is added to the introduction.

2. Pg2 Line 5, please add reference for DART applications.

   Answer: Citation to DART publication is added.

3. Starting Pg4 Line 5, the authors introduced DA in general, then at Pg 5, line 22, the authors mentioned they have implemented several flavors of the above mentioned DA schemes. It would be nice to mention which versions are implemented and which are not. Alternatively, the authors could start with the summary of DATeS 's implementation, then provide details of each implementation.

   Answer: In table 3, we provide citations to the implemented algorithms, which directly indicate the implemented flavor. We argue that it is nearly impossible to enumerate the unimplemented algorithms, since data assimilation is a rich field that is continuously growing. Moreover, detailed description of each implementation would make the manuscript unnecessary long. These details, however can be accessed through the code, and the documentation available on the package website.

4. The authors have mentioned in several places that FORTRAN and C are low-level languages. They are not. Low-level languages are generally referred to machine language and assembly language.

   Answer: To avoid the confusion related to low- and high-level languages, we opted to refer to such languages as high-performance languages. We updated the text accordingly.

5. The authors are encouraged to develop a Graphic User Interface (GUI) for flexibility and ease of use.

   Answer: We agree that a GUI would be useful for experimental purposes, however this is beyond the scope of this version of the package. We will definitely consider it in future releases.

6. The DA software and programming language comparison on pg 2 can be formatted better in terns of the software cost, learning curve, ease of access, extensibility etc. to bring out the "unified" nature of the Python implementation.

   Answer: We modified the text slightly to highlight and discuss these issues. However, the discussion in this section is not meant as a general detailed discussion, or to promote Python over other lower level languages. We focused on languages used in the data assimilation research, and we understand that personal preference and computational resources together play an essential role in deciding which language to use. DATeS is meant to bring together pieces written in lower level languages, such as C and Fortran, and enable proper interaction through proper interfaces written in Python.

7. The authors mentioned to compare different DA schemes, it would be nice to include an example of DA results comparison with DATeS .

   Answer: We agree that adding some sort of comparison made using DATeS is useful. We opted to cite work that presents numerical comparisons between

DA schemes carried out using DATeS framework instead of reproducing results. Moreover, we added a section on benchmarking which provides another form of comparison that can be done using DATeS .

8. I also agree with another referee that limitations of DATeS should be pointed out.

Answer: A discussion is added to the discussion and conclusion (Section 6).

---

## Author Response (AR2)

**DATeS: A Highly-Extensible Data Assimilation Testing Suite v1.0**

Ahmed Attia and Adrian Sandu

Dear Editor,

Please find attached the marked-up revised version of the manuscript "DATeS: A Highly-Extensible Data Assimilation Testing Suite v1.0" by A. Attia and A. Sandu.

We thank the reviewers for their useful feedback. A point-wise list of responses to each of the reviewers' comments is attached.

Regards,
Ahmed Attia

**Referee #2; Kody Law:**

The paper is substantially improved, and it can be published once the other referees are satisfied. Answer: Thank you.

1. Note equation (10) is not correct and should be corrected. Answer: We kindly argue that the equation itself is correct, however we agree that there is some confusion caused by the text following that equation. This issue is now addressed in the updated version of the manuscript; see page 19.

2. A mathematical definition of rank histogram should be given, for the results shown, since there is not a unique way to define it. Answer: We value this suggestion, however, to keep the discussion simple and to avoid any confusion, we refer the readers to another article which presents a mathematical description and a detailed discussion on the interpretation and usefulness of rank histograms. The text has been updated accordingly; see page 20.